# MULTIGRAPH MESSAGE PASSING WITH BI-DIRECTIONAL MULTI-EDGE AGGREGATIONS

## ABSTRACT

Graph Neural Networks (GNNs) have seen significant advances in recent years, yet their application to multigraphs, where parallel edges exist between the same pair of nodes, remains under-explored. Standard GNNs, designed for simple graphs, compute node representations by combining all connected edges at once, without distinguishing between edges from different neighbors. There are some GNN architectures proposed specifically for multigraph tasks, yet these architectures perform only node-level aggregation in their message-passing layers, which limits their expressive power. Furthermore, these approaches either lack permutation equivariance when a strict total edge ordering is absent, or fail to preserve the topological structure of the multigraph. To address all these shortcomings, we propose MEGA-GNN, a unified framework for message passing on multigraphs that can effectively perform diverse graph learning tasks. Our approach introduces a two-stage aggregation process in the message passing layers: first, parallel edges are aggregated, followed by a node-level aggregation that operates on aggregated messages from distinct neighbors. We show that MEGA-GNN supports permutation equivariance and invariance properties. We also show that MEGA-GNN is universal given a strict total order on the edges. Experiments on synthetic and real-world financial transaction datasets demonstrate that MEGA-GNN either significantly outperforms or is on par with the accuracy of state-of-the-art solutions.

## 1 INTRODUCTION

Graph Neural Networks (GNNs) (Xu et al. (2019); Gilmer et al. (2017); Veličković et al. (2018); Corso et al. (2020); Hamilton et al. (2017)) have become Swiss Army knives for learning on graph-structured data. However, their widespread adoption has primarily focused on simple graphs, where only a single edge can connect a given pair of nodes. This simplification overlooks a crucial aspect of many real-world scenarios, where multigraphs, graphs that feature parallel edges between the same pair of nodes, are common. For instance, financial transaction networks, communication networks and transportation systems are often modeled as multigraphs, allowing multiple different interactions between the same two nodes. Existing GNN architectures are inherently limited in handling such multigraph structures since they aggregate messages only at the node level, combining information from all incoming edges at once without distinguishing between parallel edges and their sources. Although such an aggregation mechanism is sufficient to construct provably powerful GNN models on simple graphs (Maron et al. (2019); Xu et al. (2019)), the expressivity of such models are theoretically and empirically limited on directed multigraphs as shown by Egressy et al. (2024).

In the literature, only two GNN-based solutions, namely Multi-GNN (Egressy et al. (2024)) and ADAMM (Sotiropoulos et al. (2023)), have been proposed for multigraphs, yet both exhibit critical limitations as summarized in Table 1. Multi-GNN, while addressing expressivity concerns, fails to preserve permutation equivariance. ADAMM, on the other hand, fails to effectively support diverse graph learning tasks, thus limiting its practical applicability. To address these limitations, we propose **M**ulti-**E**d**G**e **A**ggregation GNNs, henceforth referred to as **MEGA-GNN**. MEGA-GNN is a unified message passing framework specifically designed for multigraphs. It employs a two-stage message aggregation process: first, parallel edges between the same two nodes are aggregated, and then, the aggregated messages from distinct neighbors are further combined at the node level.

Table 1: Closely-related work vs. our method. MP stands for message passing.

| Properties & Capabilities | Multi-GNN | ADAMM | MEGA-GNN (ours) |
|---|:---:|:---:|:---:|
| Proof of Universality | ✓ | | ✓ |
| Bi-directional MP | ✓ | | ✓ |
| Edge Embeddings | ✓ | | ✓ |
| Node Embeddings | ✓ | ✓ | ✓ |
| Permutation Equivariance | | ✓ | ✓ |
| Multi-Edge Aggregations | | ✓ | ✓ |
| Multi-Edge Aggregations in MP | | | ✓ |

Our theoretical analyses show that MEGA-GNN is provably powerful, meaning it can detect any directed subgraph pattern within multi-graphs if a strict total ordering of the edges is possible.

Our main contributions are three fold: **(1)** We introduce a generic bi-directional message passing framework for multigraphs that incorporates artificial nodes between pairs of nodes to aggregate parallel edges before message passing, which makes it effective across diverse multigraph learning tasks. **(2)** We prove that our framework supports essential properties such as permutation equivariance and universality. **(3)** We validate our framework on financial transaction datasets, covering detection of illicit transactions and phishing accounts. We outperform the state-of-the-art for illicit transaction detection and match the state-of-the-art results for phishing account detection.

## 2 LIMITATIONS OF THE EXISTING SOLUTIONS

In the literature, two key works specifically address multigraphs: Multi-GNN Egressy et al. (2024) and ADAMM Sotiropoulos et al. (2023).

**Multi-GNN** introduced a provably powerful GNN architecture with simple adaptations for directed multigraphs. A notable contribution of Multi-GNN was the multigraph port ID assignment on edges, which allows the model to differentiate between edges from the same neighbor and edges from different neighbors, an essential feature for handling multigraph structures (see Figure 1). In addition to multigraph port numbering, Multi-GNN incorporates ego-IDs (You et al. (2021)) and bi-directional message passing (Jaume et al. (2019)). The authors also proved that with these three adaptations, it is possible to assign unique node IDs in connected directed multigraphs, making their solution universal.

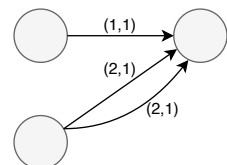

Figure 1: Directed multigraph port numbering of Multi-GNN (Egressy et al. (2024)).

The main drawback of Multi-GNN is that augmenting edge features with multigraph port numbers leads to the loss of permutation invariance and equivariance properties (see Proposition 2.1 and the proof provided in Appendix A.1). These properties are critical for many graph learning tasks to ensure that the model's predictions remain consistent under arbitrary node or edge permutations.

**Proposition 2.1.** *The multigraph port numbering scheme of Egressy et al. (2024), is not permutation-equivariant in the absence of a contextually-driven strict total ordering of edges.*

**ADAMM** proposed aggregating all parallel edges between two nodes to a single undirected super-edge, thereby transforming the multigraph into a simple graph before message passing. An initial features for this super-edge is computed using DeepSet and the subsequent message passing layers operate on this features. However, this approach results in a loss of critical structural information inherent in the multigraph. Specifically, by collapsing multiple edges into a single one, the formulation can not generate features for individual edges. Consequently, it cannot perform graph learning tasks that require edge features.

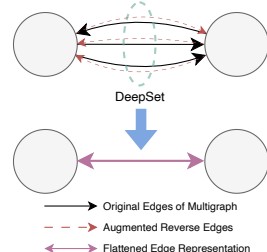

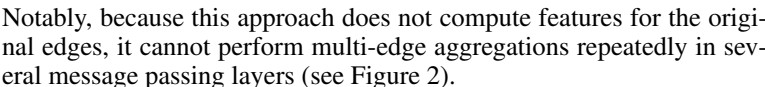

Figure 2: Illustration of ADAMM Sotiropoulos et al. (2023)

Notably, because this approach does not compute features for the original edges, it cannot perform multi-edge aggregations repeatedly in several message passing layers (see Figure 2).

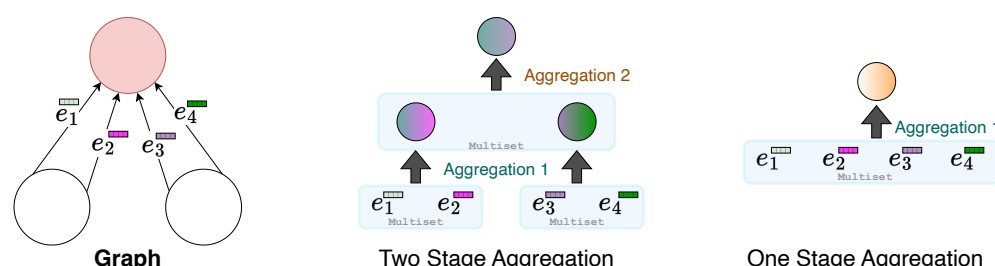

Figure 3: **Comparison of two-stage and single-stage aggregation strategies for multigraphs**. The left panel illustrates a sample multi-graph. The center panel depicts the two-stage aggregation process, wherein the first stage aggregates parallel edges, followed by a second stage node-level aggregation. The right panel shows the single-stage aggregation scheme commonly employed by standard GNNs, which would aggregate all edges, including the parallel edges, at once.

Another major limitation of ADAMM is its lack of support for bi-directional message passing, which has been shown to lead to significant accuracy gains for directed multigraphs(Egressy et al. (2024)).

**In summary**, a general framework for message passing on multigraphs that supports permutation invariance and a general set of graph learning tasks has yet to be proposed. This paper addresses this gap by introducing a novel message passing architecture that is explicitly designed for multigraphs.

## 3 MULTIGRAPH MESSAGE PASSING

This section introduces a novel message-passing scheme for multigraphs which combines node-level aggregation with *multi-edge aggregation* and supports a general set of graph learning tasks such as node, edge, and graph classification. Additionally, the resulting models are permutation invariant or equivariant depending on the chosen multi-edge aggregation function and the downstream task.

### 3.1 MOTIVATION FOR MULTI-EDGE AGGREGATIONS IN MULTIGRAPH GNNS

We argue that a two-stage aggregation approach is more powerful than a single-stage aggregation in a multigraph setting, wherein aggregation of the parallel edges between the same source and destination nodes is performed before a node-level aggregation. We first give a motivating example, which shows that two-stage aggregation functions are more expressive than their single-stage counterparts.

Consider the example graph depicted in Figure 3, with four edges $e_1, e_2, e_3$ and $e_4$. Suppose that we have an aggregation function that operates on multisets and can simultaneously compute both the sum and the maximum of the elements included in the set. To demonstrate that two-stage aggregation approach is more powerful than the single-stage one, we will show that two-stage approach can perform any computation that the single-stage approach can, but not vice versa.

A single-stage scheme, can compute only $(\text{SUM}\{\!\!\{e_1, e_2, e_3, e_4\}\!\!\}$ and $\text{MAX}\{\!\!\{e_1, e_2, e_3, e_4\}\!\!\})$. The two-stage approach can also compute these results by aggregating the edges in pairs first: $(\text{SUM}\{\!\!\{\text{SUM}\{\!\!\{e_1, e_2\}\!\!\}, \text{SUM}\{\!\!\{e_3, e_4\}\!\!\}\}\!\!\}, \text{MAX}\{\!\!\{\text{MAX}\{\!\!\{e_1, e_2\}\!\!\}, \text{MAX}\{\!\!\{e_3, e_4\}\!\!\}\}\!\!\})$. However, the two-stage aggregation allows for more nuanced computations that the single-stage approach cannot perform. For example, the two-stage approach can compute $\text{SUM}\{\!\!\{\text{MAX}\{\!\!\{e_1, e_2\}\!\!\}, \text{MAX}\{\!\!\{e_3, e_4\}\!\!\}\}\!\!\}$ and $\text{MAX}\{\!\!\{\text{SUM}\{\!\!\{e_1, e_2\}\!\!\}, \text{SUM}\{\!\!\{e_3, e_4\}\!\!\}\}\!\!\}$, which the single-stage approach cannot compute.

This distinction is significant; the two-stage aggregation enables us to capture the statistics of edges originating from different neighbors. Consider a financial application, in which a receiver may receive the highest total payment from one sender (Sender 1), while the highest single payment may originate from another sender (Sender 2). Using the two-stage scheme, we can detect the maximum of the sums of payments from each sender, allowing us to distinguish between their contributions. In contrast, the single-stage scheme can only compute the maximum among all payments without differentiating between identical and distinct senders, which can obscure critical insights.

## 3.2 PRELIMINARIES

**Notation**: We denote multisets with $\{\{\}\}$ and sets with $\{\}$. Let $\mathcal{G} = (\mathcal{V}, \mathcal{E})$ be a directed multigraph with $|\mathcal{V}| = v$ nodes and $|\mathcal{E}| = e$ edges. The edges $\mathcal{E} = \{\{e_{ij} = (i, j) \mid i, j \in \mathcal{V}\}\}$ form an ordered multi-set of pairs of nodes, where each pair $(i, j)$ represents a directed edge from node $i$ to node $j$. In a multigraph, several edges may exists between between the same source and destination nodes, which we refer to as parallel edges. The incoming and outgoing neighbors of node $i \in \mathcal{V}$ are denoted by $N_{in}(i) = \{\{j \in \mathcal{V} | (j, i) \in \mathcal{E}\}\}$ and $N_{out}(i) = \{\{j \in \mathcal{V} | (i, j) \in \mathcal{E}\}\}$, respectively.

In this work, we consider attributed graphs. The nodes and the edges of the graph $\mathcal{G}$ are associated with initial feature vectors $\mathbf{x}_i^{(0)} \in \mathbb{R}^{D_n}$ and $\mathbf{e}_{ij}^{(0)} \in \mathbb{R}^{D_e}$, and latent feature vectors $\mathbf{x}_i^{(l)} \in \mathbb{R}^{D_n^h}$ and $\mathbf{e}_{ij}^{(l)} \in \mathbb{R}^{D_e^h}$, where $D_n, D_e, D_n^h$ and $D_e^h \in \mathbb{R}$ represent the dimensions of the given vectors, respectively, and $l \in [1, L]$, $L$ is the number of layers of the model.

In a multigraph, there can be parallel edges between the same two nodes, each with its own feature vector, which requires additional considerations. Let $\mathcal{E}^{supp} \subseteq \mathcal{E}$ denote the support set of $\mathcal{E}$, which consists of unique $(i, j)$ pairs that have a multiplicity of at least one in the multi-set $\mathcal{E}$. Furthermore, let $ME_{ij} = \{\{(i, j) \in \mathcal{E}\}\}$ denote the multiset of parallel edges between a source node $i$ and a destination node $j$. Assume also that the cardinality of $ME_{ij}$ is $P_{ij}$. The initial and the latent feature vectors of these parallel edges are denoted with $\mathbf{e}_{ijp}^{(0)} \in \mathbb{R}^{D_e}$ and $\mathbf{e}_{ijp}^{(l)} \in \mathbb{R}^{D_e^h}$, where $p \in \{1..P_{ij}\}$.

**Message-Passing GNNs**: In the standard Message Passing Neural Networks (MPNNs) proposed by Gilmer et al. (2017), the information on the graph is propagated using an iterative application of local message passing operations. The following shows how an MPNN can be applied to a multigraph:

$$\mathbf{x}_j^{(l)} = g_v^{(l-1)}\left(\mathbf{x}_j^{(l-1)}, \text{AGG}\{\{f^{(l-1)}(\mathbf{x}_i^{(l-1)}, \mathbf{e}_{ijp}^{(l-1)}) \mid i \in N_{in}(j), p \in \{1..P_{ij}\} \}\}\right), \quad (1)$$

where $f^{(l)}$ is a function that constructs the messages from the incoming neighbors of node $i$. These messages are then aggregated using the AGG function, which is typically a permutation-invariant operation such as sum, mean, or max. After the message aggregation step, the latent features of node $i$ is updated using the update function $g_v^{(l)}$. Additionally, latent edge features could also be updated using the following formula, where $g_e^{(l)}$ is the edge update function:

$$\mathbf{e}_{ijp}^{(l)} = g_e^{(l-1)}\left(\mathbf{x}_i^{(l-1)}, \mathbf{e}_{ijp}^{(l-1)}, \mathbf{x}_j^{(l-1)}\right). \quad (2)$$

We now formally define some key concepts that will be used throughout the paper.

**Definition 3.1. (Strict Total Order)** *A multiset of edges $\mathcal{E}$ in a graph is said to have a strict total order under a binary relation $<$ if the following conditions hold for every $a$, $b$ and $c \in \mathcal{E}$: (1) $a \neq b$, either $a < b$ or $b < a$, (2) not $a < a$, (3) if $a < b$ and $b < c$, then $a < c$ (see Munkres (2000), p. 22).*

**Definition 3.2. (Universality)** *An MPNN is universal if it can approximate every invariant or equivariant continuous functions defined on graphs (Keriven & Peyré (2019)). According to Loukas (2020), universality is achieved if the following conditions are satisfied: there are enough layers, the layers are sufficiently expressive and wide, and the nodes can uniquely distinguish each other.*

**Definition 3.3. (Permutation Equivariance)** *A function $\psi$ is said to be permutation equivariant with respect to node and edge permutations if, for any permutation $\rho$ acting on the nodes and edges of a graph $\mathcal{G} = (\mathcal{V}, \mathcal{E})$, the following holds:*

$$\psi(\rho \circ \mathcal{G}(\mathcal{V}, \mathcal{E})) = \rho \circ \psi(\mathcal{G}(\mathcal{V}, \mathcal{E}))$$

## 3.3 MULTIGRAPH MESSAGE PASSING WITH MULTI-EDGE AGGREGATIONS

Commonly used GNN architectures Kipf & Welling (2017); Veličković et al. (2018); Hamilton et al. (2017) are not designed for multigraphs. Performing message aggregation solely at the node level as shown in Equation 1, can limit the expressive power of these models on multigraph learning tasks. As a remedy, we propose a two-stage aggregation process, wherein the first stage, termed *multi-edge aggregation*, aggregates parallel edges, followed by a second stage node-level aggregation. To enable effective multi-edge aggregation, we introduce novel *artificial nodes* into the multigraph.

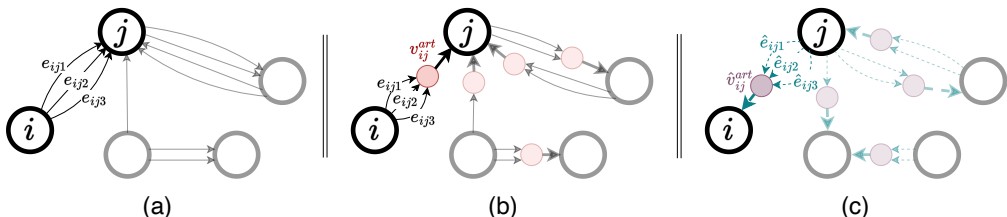

Figure 4: **Illustration of Multi-Edge Aggregation using artificial nodes in a multigraph**. (a) A multigraph with parallel edges $e_{ij1}, e_{ij2}, e_{ij3}$ between the nodes $i$ and $j$. (b) Artificial nodes positioned between adjacent pairs of nodes to handle the aggregation of parallel edges. First, the information from the *parallel* edges are aggregated into some embedding vectors in the artificial nodes, and then the destination node performs a node-level aggregation on these embeddings. (c) The reverse message passing mechanism. In directed multigraphs, reverse edges are created in the reverse direction of the original edges, and then additional artificial nodes are introduced to handle the aggregation of these reverse edges. Separate message computations are performed to handle the original and reversed edges. The result is a bi-directional multigraph message passing solution.

We define the set of *artificial nodes*, positioned between each unique pair of nodes in the multigraph as shown in Figure 4 (b) as follows:

$$\mathcal{V}^{art} = \{v_{ij}^{art} \mid (i,j) \in \mathcal{E}^{supp}\}. \tag{3}$$

The incoming neighbors of the *artificial node* $v_{ij}^{art}$ is given by the multiset $ME_{ij}$, the cardinality of which is $P_{ij}$. Let $\mathbf{h}_{ij} \in \mathbb{R}^d$ be the latent features of the *artificial node* $v_{ij}^{art}$.

The message-passing algorithm for multigraphs is adapted to perform information aggregation in two stages. First, parallel edges are aggregated at the artificial nodes, thereby reducing the multigraph to an equivalent simple graph:

$$\mathbf{h}_{ij}^{(l-1)} = \text{EdgeAgg}\{\mathbf{e}_{ijp}^{(l-1)} \mid p \in \{1..P_{ij}\}\}. \tag{4}$$

In the second stage, aggregation occurs at the original graph nodes:

$$
\begin{aligned}
\mathbf{a}_j^{(l-1)} &= \text{AGG}\{f^{(l-1)}(\mathbf{x}_i^{(l-1)}, \mathbf{h}_{ij}^{(l-1)}) \mid i \in N_{in}(j)\} \\
\mathbf{x}_j^{(l)} &= g_v^{(l-1)}\left(\mathbf{x}_j^{(l-1)}, \mathbf{a}_j^{(l-1)}\right),
\end{aligned}
\tag{5}
$$

where $N_{in}(j) = \{i \in \mathcal{V} \mid (i,j) \in \mathcal{E}^{supp}\}$, which is not a multiset, it only includes distinct incoming neighbors of node $j$. As in 2, the latent features of each individual edge can be updated without loss of generality:

$$\mathbf{e}_{ijp}^{(l)} = g_e^{(l-1)}\left(\mathbf{x}_i^{(l-1)}, \mathbf{e}_{ijp}^{(l-1)}, \mathbf{h}_{ij}^{(l-1)}\right). \tag{6}$$

This two-stage aggregation process, enabled by our innovative use of artificial nodes, preserves the topology of the original multigraph. Equations 4, 5, and 6 can be applied iteratively, allowing for effective propagation and updates of both node and edge latent features in each message-passing layer. Unlike previous methods such as ADAMM Sotiropoulos et al. (2023), which collapses parallel edges into a single edge prior to message passing—thereby preventing updates on the original multigraph edges—our approach maintains distinct edges through individual updates, enabling richer feature propagation in multigraph learning tasks.

### 3.4 BI-DIRECTIONAL MULTIGRAPH MESSAGE PASSING

Bi-directional message passing (Egressy et al. (2024)) has been shown to significantly improve model expressivity, especially for directed graphs. In standard message passing, a node does not receive information from its outgoing neighbors unless those neighbors are also incoming. Consequently, the node is unable to account for its outgoing edges.

We ensure that our proposed model, MEGA-GNN, also supports bi-directional message passing for multigraphs. In MEGA-GNN, a node is able to receive messages from its outgoing neighbors by using the reversed versions of their outgoing edges. Formally, the reversed edge set (illustrated in Figure 4(c)) is defined as $\hat{\mathcal{E}} = \{\{\hat{e}_{ji} = (j,i) \mid (i,j) \in \mathcal{E}\}\}$, and the edge features for each reverse edge are initialized using the corresponding edge features from the original graph $\hat{\mathbf{e}}_{jip}^{(0)} = \mathbf{e}_{ijp}^{(0)}$, where $p \in \{1..P_{ij}\}$. In the reversed multigraph, *artificial nodes* $\hat{\mathcal{V}}^{art}$ are placed between each unique pair of nodes, in a manner that mirrors their positioning in the original multigraph:

$$\hat{\mathcal{V}}^{art} = \{\hat{v}_{ij}^{art} \mid (j,i) \in \mathcal{E}^{supp}\}. \tag{7}$$

The incoming neighbors of the *artificial node* $\hat{v}_{ij}^{art}$ is given by the multiset $ME_{ji}$, the cardinality of which is $P_{ji}$. Let $\hat{\mathbf{h}}_{ij} \in \mathbb{R}^d$ be the latent features of the *artificial node* $\hat{v}_{ij}^{art}$.

$$\hat{\mathbf{h}}_{ij}^{(l-1)} = \text{EdgeAgg}\{\hat{\mathbf{e}}_{jip}^{(l-1)} \mid p \in \{1..P_{ji}\}\}. \tag{8}$$

$\hat{\mathbf{h}}_{ij}^{(l)}$ is the aggregation of the incoming edges from node $i$ to node $j$ in the reversed multigraph,

$$\hat{\mathbf{a}}_{j}^{(l-1)} = \text{AGG}\{\hat{f}^{(l-1)}(\mathbf{x}_{i}^{(l-1)}, \hat{\mathbf{h}}_{ij}^{(l-1)}) \mid i \in N_{out}(j)\}$$
$$\mathbf{x}_{j}^{(l)} = g_{v}^{(l-1)}\left(\mathbf{x}_{j}^{(l-1)}, \mathbf{a}_{j}^{(l-1)}, \hat{\mathbf{a}}_{j}^{(l-1)}\right), \tag{9}$$

$N_{out}(j) = \{i \in \mathcal{V} \mid (j,i) \in \mathcal{E}^{supp}\}$ and $\hat{f}^{(l)}$ is a function that constructs the messages for the reversed edges. As in 2, the latent features of each individual edge can be updated without loss of generality:

$$\hat{\mathbf{e}}_{ijp}^{(l)} = \hat{g}_{e}^{(l-1)}\left(\mathbf{x}_{i}^{(l-1)}, \hat{\mathbf{h}}_{ij}^{(l-1)}, \mathbf{x}_{j}^{(l-1)}\right), \tag{10}$$

where $\hat{g}_{e}^{(l)}$ is the update function for the reversed edges. To compute $a_{j}^{(l-1)}$ we use the same equations given in the previous subsection.

## 3.5 PROPERTIES

In this section, we establish several key theoretical properties of MEGA-GNN and Multi-GNN, analysing and comparing their expressiveness and permutation invariance under different conditions. In particular, we show that MEGA-GNN is always permutation equivariant, a property not shared by Multi-GNN. Next, we prove that MEGA-GNN is universal given a strict total order on the edges.

**Theorem 3.1. (Permutation Equivariance).** *Given permutation-invariant EdgeAgg and AGG functions, the framework defined by Equations 4, 5, and 6 is permutation-equivariant.*

The proof of Theorem 3.1 is provided in Appendix A.2. For neural networks operating on graph-structured data, permutation equivariance is a crucial property, ensuring that the model's output remains consistent regardless of the ordering of nodes and edges. Our model is explicitly designed to maintain permutation equivariance at both the node and edge levels. Permutation invariance for graph-level outputs can be achieved by using a permutation-invariant *graph readout* function.

**Lemma 3.1.** *The message-passing framework defined by Equations 4, 5, and 6 can compute unique node IDs in connected multigraphs given a strict total ordering of the edges.*

The proof of Lemma 3.1 (provided in A.3) shows that, when a strict total ordering among the edges (e.g., based on timestamps or other contextual edge features) is available, the model is able to assign unique identifiers to the nodes in a connected multigraph.

**Theorem 3.2. Universality.** *Given sufficient layers with sufficient expressiveness and width as well as a strict total ordering of the edges, the MEGA-GNN is universal.*

The definition of universality is given in Definition 3.2. Lemma 3.1 establishes that the MEGA-GNN can compute unique node IDs in connected multigraphs when a contextually-driven strict total ordering of edges is provided, which makes nodes to uniquely distinguish each other. Assuming MEGA-GNN also has sufficient layers, depth and expressiveness, it meets all the conditions presented in Loukas (2020) and is, therefore, universal in the sense defined by Loukas (2020).

We emphasize that the ability to assign unique node IDs does not violate the permutation equivariance of MEGA-GNN. In practice, the model does not explicitly assign IDs or rely on node/edge indices or their order. Instead, edge features such as timestamps provide a natural strict total ordering. As shown theoretically, if such an ordering exists, MEGA-GNN can assign unique node IDs, ensuring that permutation equivariance and universality can coexist without conflict.

All the above claims remain valid when the bi-directional message passing mechanism of Section 3.4 is used. Extending Theorem 3.1 to the bi-directional case is straightforward, and the proof of Lemma 3.1 in Appendix A.3 already incorporates bi-directional message passing.

To sum up, the theoretical analyses presented in this section lead to the following insights:

- Both MEGA-GNN and Multi-GNN are permutation equivariant and universal when there is a strict total ordering of the edges.

- When such an ordering is not available, 1) MEGA-GNN is permutation equivariant but non-universal, and 2) Multi-GNN is universal but not permutation equivariant.

## 4 EXPERIMENTAL RESULTS

This section presents our experiments designed to evaluate the performance of MEGA-GNN. We focus on financial transaction graphs, which are multigraphs, and address two key applications in financial crime analysis: illicit transaction detection through edge classification and phishing account detection in Ethereum transactions through node classification. We provide detailed descriptions of the datasets used in our experiments and the baselines against which we compare our method.

**Anti-Money Laundering (AML)**: Given the concerns over privacy and legal regulations, there is a scarcity of publicly available financial transaction data. Recently, Altman et al. (2023) release realistic synthetic financial transaction data for money laundering detection. In this synthetic data, a multi-agent world simulates transactions between entities such as banks, individuals, and companies. We use Small_LI, Small_HI, Medium_LI, and Medium_HI datasets, where LI indicates low-illicit and HI indicates high-illicit, referring to the ratios of illicit transactions in the datasets. The task is to perform binary classification on the transactions, labeling them as either illicit or non-illicit.

**Ethereum Phishing Transaction Network**: Since access to real financial transaction data from banks is limited due to privacy concerns, cryptocurrencies provide an alternative data source. In our study, we use the Ethereum transaction dataset (Chen et al. (2021)), where accounts are treated as nodes and transactions as edges. Each node has a label indicating whether it is a phishing node or not. The edges contain attributes such as transaction amount and timestamp.

**Implementation**: We implement our solution using PyTorch Geometric Fey & Lenssen (2019). We use the same temporal split as the Multi-GNN baseline across the AML datasets and apply a temporal split of 65-15-20 on the ETH dataset across the edges (transactions). We adopt Ego-IDs (You et al. (2021)) in our model in the edge classification experiments (see Table 2) and incorporate bi-directional message passing (see Section 3.4 for details) in both edge and node classification experiments (see Table 2 and 3). To ensure the statistical significance of our results, each experiment is repeated five times with different random seeds, and the mean and the standard deviation of these runs are reported. Further details about the models and the hyperparameters are provided in Appendix B.2 and Appendix B.1, respectively.

**Baselines** We select several baselines for comparison. In particular, we include GFP+LightGBM and GFP+XGBoost Blanuša et al. (2024), as well as Multi-GNN Egressy et al. (2024), as the state-of-the-art approaches. We should note that the baseline method Multi-GNN also incorporates Ego-IDs (You et al. (2021)). Additionally, within the general multigraph message passing framework (see Section 3.3), we explore different aggregation methods such as GIN (Xu et al. (2019)), PNA (Corso et al. (2020)) and GenAgg (Kortvelesy et al. (2023)), can be applied to both multi-edge aggregation (see Appendix B.2 for details) and node-level aggregation. Note that our models can also use a combination of different multi-edge and node aggregation functions.

Table 2: Minority-class F1 scores (%) on AML edge classification task. GIN and PNA baselines are designed for simple graphs. GFP performs a combinatorial search to extract discriminative subgraph patterns from multigraphs. Multi-GNNs are specifically designed to mine patterns in multigraphs.

| Method | AML Small HI | AML Small LI | AML Medium HI | AML Medium LI |
|---|---|---|---|---|
| GIN Xu et al. (2019) | 28.70 ± 1.13 | 7.90 ± 2.78 | 42.20 ± 0.44 | 3.86 ± 3.62 |
| PNA Corso et al. (2020) | 56.77 ± 2.41 | 14.85 ± 1.46 | 59.71 ± 1.91 | 27.73 ± 1.65 |
| GFP+LightGBM Blanuša et al. (2024) | 62.86 ± 0.25 | 20.83 ± 1.50 | 59.48 ± 0.15 | 24.74 ± 0.46 |
| GFP+XGBoost Blanuša et al. (2024) | 63.23 ± 0.17 | 27.28 ± 0.69 | 65.70 ± 0.26 | 31.03 ± 0.22 |
| Multi-GIN Egressy et al. (2024) | 64.79 ± 1.22 | 26.88 ± 6.63 | 58.92 ± 1.83 | 16.30 ± 4.73 |
| Multi-PNA Egressy et al. (2024) | 68.16 ± 2.65 | 33.07 ± 2.63 | 66.48 ± 1.63 | 36.07 ± 1.17 |
| Multi-GenAgg | 64.92 ± 3.85 | 36.35 ± 4.07 | 66.44 ± 1.30 | 37.72 ± 0.72 |
| MEGA-GIN | 70.83 ± 2.18 | 43.66 ± 0.54 | 68.83 ± 1.66 | 39.03 ± 1.88 |
| MEGA-PNA | 74.01 ± 1.55 | 45.07 ± 2.26 | 78.26 ± 0.11 | 49.40 ± 0.54 |
| MEGA-GenAgg | 74.88 ± 0.38 | 46.29 ± 0.41 | 76.69 ± 0.30 | 44.89 ± 0.06 |
| MEGA(PNA)-GIN | 74.69 ± 0.50 | 41.36 ± 0.73 | 72.63 ± 0.60 | 42.23 ± 0.24 |
| MEGA(GenAgg)-GIN | 72.24 ± 3.42 | 45.73 ± 0.47 | 71.34 ± 1.78 | 45.33 ± 2.08 |
| MEGA(GenAgg)-PNA | 71.95 ± 1.67 | 41.94 ± 2.02 | 76.90 ± 0.70 | 47.69 ± 0.48 |

Table 3: Minority-class F1 scores (%) on ETH node classification task.

| Method | F1 |
|---|---|
| ADAMM-GIN Sotiropoulos et al. (2023) | 34.73 ± 15.75 |
| ADAMM-PNA Sotiropoulos et al. (2023) | 37.99 ± 5.41 |
| ADAMM-GenAgg Sotiropoulos et al. (2023) | 33.10 ± 19.32 |
| Multi-GIN Egressy et al. (2024) | 51.34 ± 3.92 |
| Multi-PNA Egressy et al. (2024) | 64.61 ± 1.40 |
| Multi-GenAgg | 44.60 ± 21.50 |
| MEGA-GIN | 57.45 ± 1.14 |
| MEGA-PNA | 64.84 ± 1.73 |
| MEGA-GenAgg | 61.12 ± 2.55 |

Table 4: Ablation of MEGA-GNN variants. MP stands for message passing. We highlight the best two results.

| Ablation | Small_LI | Small_HI | ETH-Kaggle |
|---|---|---|---|
| MEGA-GIN w/ Ego-IDs & Bi-directional MP | 43.66 ± 0.54 | 70.83 ± 2.18 | 55.19 ± 2.33 |
| MEGA-GIN w/ Bi-directional MP | 41.67 ± 1.51 | 72.50 ± 3.26 | 57.45 ± 1.14 |
| MEGA-GIN w/ Ego-IDs | 40.79 ± 1.91 | 69.59 ± 1.07 | 42.82 ± 3.34 |
| MEGA-GIN (Unidirectional MP) | 41.45 ± 2.13 | 69.98 ± 2.02 | 43.56 ± 2.67 |
| MEGA-PNA w/ Ego-IDs & Bi-directional MP | 45.07 ± 2.26 | 74.01 ± 1.55 | 60.02 ± 5.10 |
| MEGA-PNA w/ Bi-directional MP | 44.70 ± 1.20 | 74.98 ± 1.59 | 64.84 ± 1.73 |
| MEGA-PNA w/ Ego-IDs | 45.61 ± 0.36 | 73.61 ± 0.55 | 57.62 ± 1.23 |
| MEGA-PNA (Unidirectional MP) | 43.59 ± 1.79 | 73.65 ± 0.36 | 59.13 ± 0.51 |

## 4.1 EDGE CLASSIFICATION

We evaluate our proposed MEGA-GNN models across four different synthetic AML datasets, as shown in Table 2. Six distinct MEGA-GNN models are employed, using different combinations of aggregation functions across the two stages. For clarity, we adopt the following naming convention: MEGA-GIN refers to the model that applies GIN Xu et al. (2019) aggregation at both stages, while MEGA(GenAgg)-GIN applies GenAgg (Kortvelesy et al. (2023)) for multi-edge aggregation and GIN for node-level aggregation. The results consistently prove the effectiveness of our multi-edge aggregation mechanism in improving edge classification performance across datasets with varying sizes and levels of class imbalance, regardless of the aggregation function combinations used.

Table 2 shows that our MEGA-GNN models consistently outperform both traditional GNN baselines, such as GIN and PNA, and state-of-the-art models, such as Multi-GNN and those based on GFP, yielding significant performance benefits across all datasets. On average, our models achieve a 9.25% increase in minority-class F1 score on the HI datasets and a 13.31% improvement on the LI datasets over the state-of-the-art. A clear trend of enhanced performance with the incorporation of multi-edge aggregation is visible across all these datasets. For instance, on the AML Medium HI dataset, MEGA-GIN boosts performance from 58.92% (Multi-GIN) to 68.83%, and similarly, MEGA-PNA advances the performance from 66.48% (Multi-PNA) to 78.26%. These gains of nearly 10% are consistent across all the AML datasets, underscoring the power of multi-edge aggregation.

## 4.2 NODE CLASSIFICATION

We further evaluate our proposed models on real-world financial transactions using the ETH dataset Chen et al. (2021), which focuses on the identification of phishing accounts. The results, summarized in Table 3, highlight the effectiveness of our MEGA-GNN variants. Notably, MEGA-PNA achieves state-of-the-art performance, surpassing the Multi-PNA model Egressy et al. (2024) in terms of the minority class F1 score.

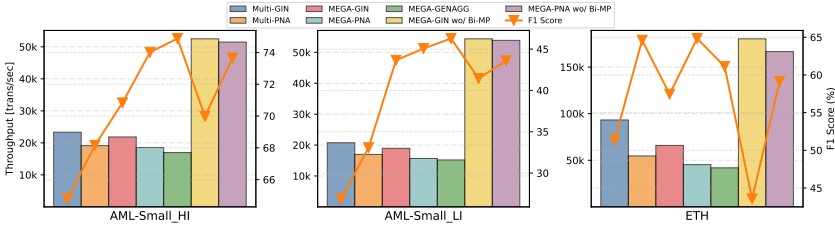

Figure 5: Throughput (transactions/sec) and minority-class F1 score (right) comparisons between MEGA-GNN and Multi-GNN variants across AML-Small_HI, AML-Small_LI, and ETH datasets.

Our method demonstrates consistent improvements over baseline models. For instance, MEGA-GIN improves the minority class F1 score by $6.11\%$ over the Multi-GIN model, while MEGA-GenAgg surpasses GenAgg-based GNN by $16.52\%$. The best-performing model, MEGA-PNA, delivers the highest F1 score among all models, achieving a slight improvement over Multi-PNA.

When compared to the ADAMM (Sotiropoulos et al. (2023)) baseline, our MEGA-GNN variants exhibit a striking improvement, consistently delivering over $20\%$ higher performance across base architectures (GIN, PNA, GenAgg). These results highlight the power of our multi-edge aggregation mechanism as well as the bi-directional message passing capabilities we contributed in Section 3.4.

### 4.3 ABLATIONS AND THROUGHPUT ANALYSIS

In Table 4, we report the results of our ablation study on the impact of EGO-IDs (You et al. (2021)) and Bi-directional Message Passing (MP) (3.4) across the Small_LI, Small_HI, and ETH-Kaggle datasets. EGO-IDs assign unique identifiers to root nodes in the sampled batch, while Bi-directional MP uses the formulation explained in Section 3.4. The unidirectional multi-edge aggregation incorporates only the two-stage aggregation mechanism described in Section 3.3.

Notably, this analysis shows that MEGA-GNN outperforms most baselines even without using bi-directional message passing and EGO-IDs. Bi-directional MP slightly improves performance for the AML datasets and significantly improves it for the ETH-Kaggle dataset, highlighting the benefit of this optimization for directed multigraphs. EGO-IDs do not yield consistent improvements for AML datasets but cause a performance degradation for the ETH-Kaggle dataset.

Figure 5 compares the inference performance of different MEGA-GNN and Multi-GNN variants in terms of the number of transactions processed per second using an NVIDIA GeForce RTX 4090 GPU. These results show that the runtime overhead of the multi-stage aggregations performed by MEGA-GNN is minimal. However, bi-directional MP leads to a significant overhead. Interestingly, MEGA-PNA without bi-directional MP offers some favorable speed and accuracy combinations.

Further discussion on the model's scalability and memory overhead is provided in Appendix D.

## 5 RELATED WORK

The expressive power of GNNs is crucial in evaluating their capabilities, and is mainly discussed in the context of simple graphs, which can have only a single edge between two graph nodes. Xu et al. (2019) demonstrated that the expressive power of standard message-passing GNNs on simple graphs is fundamentally limited by the 1-WL (Weisfeiler-Lehman) test for distinguishing isomorphic graphs. To address this limitation, they introduced the Graph Isomorphism Network (GIN), which is provably the most expressive among standard message-passing GNNs. Recent research on enhancing GNN expressivity has pursued several directions. Higher-order GNNs were proposed by Maron et al. (2019); Morris et al. (2019), Bouritsas et al. (2023); Barceló et al. (2021), leverage pre-computed local structural features around target nodes, while Bevilacqua et al. (2022; 2024); Frasca et al. (2022) focus on increasing expressivity through subgraph-based aggregation techniques.

The expressiveness of GNNs is closely linked to their universality, as more expressive models can approximate a broader class of functions. Sato et al. (2021); Loukas (2020) demonstrated that incorporating unique node identifiers can make GNNs universal, though this comes at the cost of losing

permutation invariance. However, Abboud et al. (2021) showed that partially randomized node features can achieve universality while preserving permutation invariance. In a related line of work, set functions, such as those proposed in Zaheer et al. (2017), have been shown to be universal under certain input constraints, and Fuchs* & Veličković* (2023) made the connection between universal set and graph functions. While not theoretically more expressive than GIN, Corso et al. (2020) demonstrated that combining arbitrary aggregation functions leads to better empirical performance. In Kortvelesy et al. (2023), a learnable aggregation functions were proposed that improve sample efficiency compared to both PNA and DeepSet-based aggregations on multiset neighborhoods.

There has been relatively little work on GNNs for multigraphs. Two notable prior works are Sotiropoulos et al. (2023) and Egressy et al. (2024). The former (ADAMM) transforms a multigraph into a simple graph using a DeepSet-based approach Zaheer et al. (2017), while the latter (Multi-GNN) uses simple adaptations on top of baseline GNN methods to achieve universality. However, these methods still simplify the graph structure or lose permutation equivariance.

Multi-level message aggregation schemes have been proposed also in other settings, but these methods are not directly applicable to multigraphs. For instance, Hypergraph GNNs (Feng et al. (2019); Huang & Yang (2021)) first aggregate node features within each hyperedge to compute latent features of the hyperedges, which are then used to aggregate node features. In contrast, our multigraph GNNs directly handle multi-edges, representing multiple connections between the same pair of nodes. Furthermore, our work differs from Relational GNNs (Schlichtkrull et al. (2018); Vashishth et al. (2020)), which are designed for heterogeneous graphs with different edge types. MEGA-GNNs do not require explicit edge type definitions and operate directly on edge features.

## 6 DISCUSSION AND CONCLUSION

We contribute the first message-passing framework explicitly designed for multigraphs, enabling a two-stage aggregation scheme in every message-passing layer. Unlike the method of Egressy et al. (2024), our framework guarantees permutation equivariance without strict total ordering of the edges. Our framework also achieves universality when we make the same assumptions made by Egressy et al. (2024). Furthermore, unlike the method of Sotiropoulos et al. (2023), we do not transform a multigraph into a simple graph before the message passing layers. Thus, our message passing framework supports a more general set of multigraph learning tasks including the classification of multigraph edges.

Our experiments on financial transaction datasets show that MEGA-GNN significantly outperforms or matches the accuracy of state-of-the-art multigraph GNN models. Notably, when detecting illicit transactions on the synthetic AML datasets of Altman et al. (2023), we achieve improvements of up to $13.31\%$ in minority-class F1 scores over the method of Egressy et al. (2024), demonstrating the effectiveness of MEGA-GNN. When detecting phishing accounts on the Ethereum transaction dataset of (Chen et al. (2021)), we achieve improvements of around $20\%$ in minority-class F1 scores over the method of Sotiropoulos et al. (2023), while slightly improving the results of Egressy et al. (2024).

**Limitations and Future Work**. Our work does not consider dynamically evolving multigraphs with several different snapshots. Future work could investigate strategies to support such setups. Other interesting research directions include improving the scalability and GPU performance of our methods, and building federated multigraph learning solutions with differential privacy guarantees.

## 7 REPRODUCIBILITY STATEMENT

To ensure the reproducibility of our research, we will open-source our codes in a publicly accessible repository. The experiments conducted as part of this work used only publicly available datasets.

## 8 ETHICS STATEMENT

Our work supports the global fight against financial crime by contributing provably powerful GNN-based methods for analysing financial transaction networks. Moreover, the datasets used in our

experiments do not involve any personally identifiable information. As a result, our work does not lead to any discrimination/bias/fairness concerns or privacy, security, and legal compliance issues.

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

# A APPENDIX

## A.1 PROOF OF PROPOSITION 2.1

Let $\mathcal{G} = (\mathcal{V}, \mathcal{E})$ be a multigraph, the nodes of which are associated with feature vectors $\mathbf{x}_i \in \mathbb{R}^D$ and $\mathbf{e}_{ij} \in \mathbb{R}^K$. Assume that each edge $e \in \mathcal{E}$ is assigned a port number $p(e)$ by a given port numbering scheme, and these port numbers are used as edge features as proposed by Egressy et al. (2024).

Consider a scenario where there is no inherent or contextually-driven ordering of the edges (e.g., no timestamps or other distinguishing features that naturally dictate how the ports are numbered). As a result, the assignment of port numbers to edges is arbitrary. For a node $v$ with $d$ incoming edges, there are $d!$ possible ways to assign port numbers to the incoming edges, corresponding to all possible permutations.

The latent features of the node $i$ at layer $l$ is computed as follows, where $\phi$ is the message function, and $p(j, i)$ is the port number assigned to the edge between node $j$ and node $i$"

$$\mathbf{x}_i^{(l)} = \sum_{j \in N(i)} \phi\big(\mathbf{x}_j^{(l-1)}, [\mathbf{e}_{ji}^{(l-1)} \, || \, p(j,i)]\big). \tag{11}$$

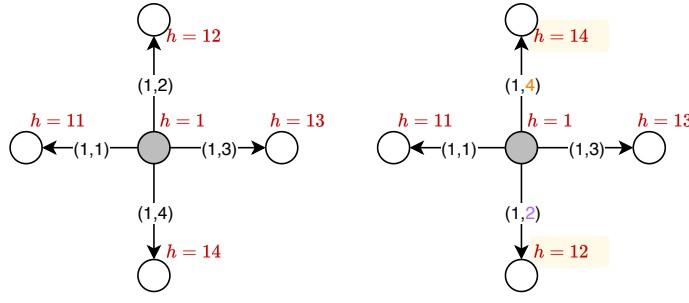

Figure 6: Counterexample for the permutation-equivariance of Multi-GNN Egressy et al. (2024). The left panel shows the graph $\mathcal{G}$ with one permutation of the port numbering, while the right panel illustrates a different permutation of the assigned port numbers. Notice how the node embeddings change when computed using Algorithm 1 from Egressy et al. (2024), highlighting that different port number assignments lead to different embeddings, thus violating permutation-equivariance

.

Now, assume that the GNN with port numbering is permutation equivariant. If our assumption is correct, permuting the port numbers assigned to edges will not affect the latent node features. Let $\sigma$ be a permutation of the port numbers. Applying this permutation to the port numbers of the edges yields a new port assignment $p_\sigma(e)$. Under this permutation, the latent node features would be:

$$\hat{\mathbf{x}}_i^{(l)} = \sum_{j \in N(i)} \phi\big(\mathbf{x}_j^{(l-1)}, [\mathbf{e}_{ji}^{(l-1)} \,||\, p_\sigma(j,i)]\big). \tag{12}$$

Since we are assuming the GNN is permutation-equivariant, the latent node features should remain the same after permuting the port numbers:

$$\mathbf{x}_i^{(l)}(p) = \hat{\mathbf{x}}_i^{(l)}(p_\sigma) \tag{13}$$

This implies that for all possible permutations $\sigma$, the latent features of the nodes should be invariant. Let the GNN, $f$, mimics the Algorithm 1 from Egressy et al. (2024), which computes unique node IDs based on port numbers. If we apply this GNN, $f$, to a *Star Graph* with order $n > 3$ with different port number permutations, resulting latent node features must be distinct. Formally, applying $f$ to a the graph with port numbering $p$ gives the latent node features $\mathbf{X}^{(1)}$, and applying $f$ to the graph with a permuted port numbering $p_\sigma$, gives a different set of node embeddings $\hat{\mathbf{X}}^{(l)}$:

$$\mathbf{X}^{(l)} = f(\mathcal{G}(\mathcal{X}, p)) \neq f(\mathcal{G}(\mathcal{X}, p_\sigma)) = \hat{\mathbf{X}}^{(l)}. \tag{14}$$

This result directly contradicts our earlier assumption that the model is permutation-equivariant with respect to node ordering, preserves the identity of node outputs. Specifically, if the GNN were permutation-equivariant, we would expect $\mathbf{X}^{(l)} = \hat{\mathbf{X}}^{(l)}$, but instead the permutation of port numbers leads to different output. Illustration of this contradictory example is given in Figure 6.

Therefore, we conclude that the arbitrary assignment of port numbers is not permutation-equivariant with respect to node orderings.

A.2 PROOF OF THEOREM 3.1

We adopt the notation and terminology introduced in Section 3.3 of this paper to ensure consistency and ease of reference.

*Proof:* The proposed message passing layer performs two aggregations over the neighborhood of a target node $j$. The first is the multi-edge aggregation, in which the latent features of the parallel edges are aggregated at artificial nodes. The one-hop neighborhood of artificial node $v_{ij}^{art}$ is given by the multiset $ME_{ij}$, the cardinality of which is $P_{ij}$:

$$\mathbf{E}_{ME_{ij}} = \{\mathbf{e}_{ijp} \,|\, p \in \{1..P_{ij}\}\}. \tag{15}$$

Let's define a permutation-invariant multi-edge aggregation function $\psi$, which operates on the neighborhood of an artificial node (the parallel edges):

$$\mathbf{h}_{ji} = \psi(\mathbf{E}_{ME_{ij}}) \tag{16}$$

Since $\psi$ is a permutation-invariant function, for any permutation function $\rho$ acting on the neighboring edges (parallel edges) of the artificial node, we have $\psi(\rho \cdot \mathbf{E}_{ME_{ij}}) = \rho \cdot \psi(\mathbf{E}_{ME_{ij}})$.

The second aggregation is then performed over the neighborhood of the target nodes, all of which happen to be artificial nodes associated with distinct neighbors in the original graph (see Figure 4).

$$\mathbf{X}_{N_{in}(j)} = \{\!\!\{\mathbf{h}_{ij} \mid (i,j) \in N_{in}(j))\}\!\!\}. \tag{17}$$

Similarly, let's define a permutation-invariant node-level aggregation function $\phi$, which operates over the neighborhood of the target nodes:

$$\mathbf{x}_j = \phi(\mathbf{x}_j, \mathbf{X}_{N_{in}(j)}) \tag{18}$$

Since $\phi$ is a permutation-invariant function, for any permutation function $\pi$ acting on the neighboring edges of a target node $i$, we have $\phi(\pi \cdot (\mathbf{x}_j, \mathbf{X}_{N_{in}(j)})) = \pi \cdot \phi((\mathbf{x}_j, \mathbf{X}_{N_{in}(j)}))$.

In our framework, $\psi$ performs multi-edge aggregation, handling parallel edges, while $\phi$ is responsible for node-level aggregation, combining messages from distinct neighbors of the target node. The two-stage aggregation scheme integrates $\psi$ and $\phi$ within a single message passing layer. Since the composition of permutation-invariant functions remains permutation-invariant, our message passing layer ($\psi \circ \phi$) is invariant to the permutations of neighboring nodes and edges ($\rho \circ \pi$). Unlike simple graphs, node permutations do not directly imply edge permutations due to the presence of parallel edges. Thus, we explicitly define the permutation of parallel edges, $\rho$, ensuring that our message passing layer remains permutation-invariant to both nodes and edges in the neighborhood of the target node.

Finally, as demonstrated by Bronstein et al. (2021), the composition of permutation-invariant layers ($f = \psi \circ \phi \circ \psi \circ \phi \cdots$) allows the construction of functions $f$ that are equivariant to symmetry group actions. In the multigraph domain, this symmetry group includes permutations of both nodes and edges, as node permutations do not directly induce edge permutations due to the presence of parallel edges. The overall permutation equivariance of the MEGA-GNN model follows from the fact that each permutation-invariant message passing layer operates independently on each node's neighborhood, regardless of the ordering of nodes or edges. Specifically, for any permutation $g \in \sum_n$ acting on the set of node and edge indices, the model's output satisfies $f(g \cdot X) = g \cdot f(X)$.

## A.3 Proof of Lemma 3.1

*Proof:* Given a graph $G(V, E)$ with $n$ nodes and $m$ edges, assume that there is a strict total ordering among the graph edges represented by an edge labeling function $L : E \to \mathbb{N} \cap [1, m]$, which assigns unique labels to the edges. We will prove that MEGA-GNN can compute unique node IDs under these assumptions.

Egressy et al. (2024) showed that a GNN can mimic a Breadth-First Search (BFS) algorithm to compute unique node IDs given pre-computed port IDs for the edges. We follow the same BFS-based approach and derive unique node ids without relying on pre-computed port IDs. Instead of the pre-computed port ids, we use the unique edge labels provided by $L(e)$ to guide the node ID assignment process. As in Egressy et al. (2024), we use the Universal Approximation Theorem Hornik et al. (1989) for MLPs, to avoid explicit construction of the MEGA-GNN layers. We also assume that the MEGA-GNN aggregates the multi-edges by computing their minimum, which is followed by a node-level aggregation, where an MLP is applied element-wise to the incoming messages, followed by another minimum computation.

Following the approach of Egressy et al. (2024), the node ID assignment algorithm starts from a root node (also called the ego node) and assigns IDs to all the other nodes connected to it via message passing . We are not going to reiterate the setup of the entire proof and focus on the differences. Namely, instead of pre-computing port-IDs, we use an edge labeling function, which assumes a strict total ordering among the edges based on the original edge features. In addition, the multi-edges are aggregated by selecting the edge with the minimum label.

---

**Algorithm 1** BFS Node ID Assignment

---

**Input:** Connected directed multigraph $G = (V, E)$ with $n$ nodes and $m$ edges, diameter $D$, and root node $r \in V$. Active nodes $X \subseteq V$ and finished nodes $F \subseteq V$. Edge Labeling $L : E \to [1, m]$

**Output:** Unique node IDs $h(v)$ for all $v \in V$ (in base $2n$)

1: $h(r) \leftarrow 1; \quad h(v) \leftarrow 0$ for all $v \in V \setminus \{r\}$
2: $F \leftarrow \emptyset; \quad X \leftarrow \{r\}$
3: **for** $k \leftarrow 1$ **to** $D$ **do**
4:    **for** $v \in V$ **do**
5:       **if** $v \in X$ **then**
6:          send $h(v) \parallel \min\{L((v, u))_{\text{out}}\}$ to $u \in N_{\text{out}}(v)$
7:          send $h(v) \parallel m + \min\{L((u, v))_{\text{in}}\}$ to $u \in N_{\text{in}}(v)$
8:          $F \leftarrow F \cup \{v\}; \quad X \leftarrow X \setminus \{v\}$
9:       **end if**
10:      **if** $v \notin F$ **then**
11:        **if** Incoming messages $M(v) \neq \emptyset$ **then**
12:          $h(v) \leftarrow \min\{M(v)\}$
13:          $X \leftarrow X \cup \{v\}$
14:        **end if**
15:      **end if**
16:    **end for**
17: **end for**

---

Our MEGA-GNN model, which mimics Algorithm 1, assigns ids to each node connected to the root node. What remains to be shown is that those assigned node IDs are unique. First, note that nodes at different distances from the root cannot end up with the same node ID. A node at distance $k$ will receive its first proposal in round $k$ and, therefore, it will have an ID with exactly $k + 1$ digits. Furthermore, an inductive argument shows that active nodes (nodes at the same distance) cannot have the same node IDs. Certainly, this is true at the start when $X = \{r\}$. Now assuming all active nodes from the previous round $(k - 1)$ had distinct node IDs, then the only way two active nodes (in round $k$) can have the same ID is if they accept a proposal from the same neighboring node. This is because, based on the induction hypothesis, proposals from different nodes will already differ in their first $k - 1$ digits. If two active nodes accepted a proposal from the same node, then they would have received different edge labels —a strict total ordering among the edges enables assignment of distinct edge labels. In addition, because $m$ is added to all incoming labels, incoming labels cannot be the same as the outgoing labels. Therefore the active nodes always accept unique proposals.

## B  IMPLEMENTATION DETAILS

### B.1  HYPERPARAMETERS

For each base GNN model and dataset, we utilized a distinct set of hyperparameters, as detailed in Table 5. The MEGA-GenAgg and Multi-GenAgg models employed the aggregation function proposed by Kortvelesy et al. (2023). In all experiments involving GenAgg, we adopted the default layer sizes of $(1, 2, 2, 4)$, and both the $a$ and $b$ parameters were made learnable, allowing the model to tailor the aggregation function to the specific downstream task. Additionally, for the GenAgg experiments, we applied the hyperparameters configured for GIN-based models as shown in Table 5.

For the AML dataset, the model was operated on neighborhoods constructed around the seed edges, while for the ETH dataset, the neighborhoods were selected around the seed nodes. In both datasets, we sampled 2-hop neighborhoods, selecting 100 neighbors per hop.

Table 5: Hyperparameter settings for AML and ETH

|  | GIN | | PNA | |
|---|---|---|---|---|
|  | **AML** | **ETH** | **AML** | **ETH** |
| $lr$ | 0.003 | 0.006 | 0.0008 | 0.0008 |
| $h$ | 64 | 32 | 20 | 20 |
| $bs$ | 8192 | 4096 | 8192 | 4096 |
| $do$ | 0.1 | 0.1 | 0.28 | 0.1 |
| $w\_ce1, w\_ce1$ | 1, 6.27 | 1, 6.27 | 1, 7 | 1, 3 |

## B.2 MULTI-EDGE AGGREGATION

In this section, we provide a detailed explanation of the GIN, PNA, and GenAgg base multi-edge aggregations in our study. The following aggregations are used to aggregate parallel edges between node $i$ to node $j$.

$$\mathbf{h}_{ij}^{(l)} = \text{MLP}\Big(\text{SUM}\{\mathbf{e}_{ijp}^{(l-1)} \mid p \in \{1..P_{ij}\}\}\Big). \tag{19}$$

The formulation in Equation 19 employs a sum-based aggregation over the multiset of incoming edges, then applies an MLP. While this approach is similar to DeepSet (Zaheer et al. (2017)), we have opted to use the term GIN-based multi-edge aggregation for consistency.

In the PNA aggregation we combined ['mean', 'min', 'max', 'std'] statistics in combination with scalars ['identity', 'amplification', 'attenuation'] around the neighborhood of each artificial node, $N_{in}(v_{ij}^{art}) = ME_{ij}$.

$$\mathbf{h}_{ij}^{(l)} = \text{MLP}\Big(\text{PNA}\{\mathbf{e}_{ijp}^{(l-1)} \mid p \in \{1..P_{ij}\}\}\Big). \tag{20}$$

GenAgg is a learnable permutation-invariant aggregator which is provably capable of representing all "standard" aggregators (see Kortvelesy et al. (2023) for details.) Different from DeepSet (Zaheer et al. (2017)), GenAgg is a scalar-valued functions applied element-wise rather than fully connected functions over the feature dimension.

$$\mathbf{h}_{ij}^{(l)} = \text{GenAgg}\Big(\text{SUM}\{\mathbf{e}_{ijp}^{(l-1)} \mid p \in \{1..P_{ij}\}\}\Big). \tag{21}$$

## C ADDITIONAL PERFORMANCE METRICS FOR AML RESULTS

In this section, we present additional results utilizing various metrics for the AML edge classification task on both small and medium datasets. These results are provided for future reference and illustrate the performance of our proposed method across key metrics, including F1, Precision, Recall, and the area under the precision-recall curve (PR-AUC).

Table 6: AML edge classification task results for Small datasets.

| Method | Small_LI | | | | Small_HI | | | |
|---|---|---|---|---|---|---|---|---|
|  | **F1** | **Precision** | **Recall** | **PR-AUC** | **F1** | **Precision** | **Recall** | **PR-AUC** |
| MEGA-GIN | $43,66 \pm 0.54$ | $63.95 \pm 4.29$ | $33.25 \pm 0.87$ | $34.94 \pm 2.76$ | $70.83 \pm 2.18$ | $70.11 \pm 4.23$ | $\mathbf{71.74 \pm 1.64}$ | $72.69 \pm 0.83$ |
| MEGA-PNA | $45.07 \pm 2.26$ | $62.05 \pm 9.97$ | $\mathbf{35.79 \pm 0.60}$ | $\mathbf{35.07 \pm 4.76}$ | $74.01 \pm 1.55$ | $76.90 \pm 4.05$ | $71.48 \pm 1.32$ | $73.91 \pm 1.05$ |
| MEGA-GenAgg | $\mathbf{46.29 \pm 0.41}$ | $\mathbf{66.16 \pm 1.22}$ | $35.62 \pm 0.50$ | $19.69 \pm 5.80$ | $\mathbf{74.88 \pm 0.38}$ | $\mathbf{78.28 \pm 2.46}$ | $71.14 \pm 1.72$ | $\mathbf{74.39 \pm 1.93}$ |

Table 7: AML edge classification task results for Medium datasets.

| Method | Medium_LI | | | | Medium_HI | | | |
|--------|-----------|--------|-----------|--------|-----------|--------|-----------|--------|
| | F1 | PR-AUC | Precision | Recall | F1 | PR-AUC | Precision | Recall |
| MEGA-GIN | $39.03 \pm 1.88$ | $67.32 \pm 11.24$ | $28.18 \pm 3.52$ | $12.06 \pm 14.93$ | $68.83 \pm 1.66$ | $70.97 \pm 2.82$ | $66.87 \pm 0.90$ | $68.82 \pm 3.06$ |
| MEGA-PNA | $\mathbf{49.40 \pm 0.54}$ | $\mathbf{75.74 \pm 2.75}$ | $\mathbf{36.69 \pm 0.72}$ | $27.85 \pm 19.38$ | $\mathbf{78.26 \pm 0.11}$ | $84.62 \pm 0.62$ | $\mathbf{73.07 \pm 0.39}$ | $\mathbf{75.51 \pm 3.58}$ |
| MEGA-GenAgg | $44.89 \pm 0.06$ | $72.09 \pm 0.68$ | $32.60 \pm 0.18$ | $\mathbf{36.50 \pm 0.40}$ | $76.69 \pm 0.30$ | $\mathbf{85.57 \pm 1.29}$ | $69.53 \pm 1.40$ | $73.37 \pm 4.19$ |

# D   SCALABILITY AND MEMORY CONSIDERATIONS

In the upcoming analysis, for simplicity, we omit the detailed examination of the computational complexity and memory overhead associated with the Multi-Layer Perceptron (MLP) layers. Instead, we focus on the unique components of our method, specifically the two-stage aggregation mechanism, which constitutes the primary contribution of our approach.

## D.1   SCALABILITY

Real-world graph data, particularly large and densely connected multigraphs, can pose significant challenges in terms of computational cost and memory usage. To assess the scalability of our approach, we analyze the asymptotic computational complexity of the two-stage aggregation process in comparison to only node-level aggregation methods, comparing aggregation functions commonly used in the PNA framework.

Let $\mathcal{V}$ denote the set of nodes, $\mathcal{E}$ the multiset of edges in the sampled batch, and $\mathcal{E}^{supp} \subseteq \mathcal{E}$ denote the support set of $\mathcal{E}$, which consists of unique $(i, j)$ pairs that have a multiplicity of at least one in the multi-set $\mathcal{E}$. Let denote the dimensionality of edge embeddings by $d$. The computational cost of performing a single-stage aggregation (for functions such as sum, standard deviation, max, and min) is $O(|\mathcal{E}|d)$.

In our two-stage aggregation process, the complexity can be broken down as follows:

- **Parallel Edge Aggregation:** The cost of aggregating parallel edges is $O(|\mathcal{E}|d)$ for all considered functions (sum, std, min, max).

- **Distinct Neighbor Aggregation:** The cost of aggregating over distinct neighbors is $O(|\mathcal{E}^{supp}|d)$, where $|\mathcal{E}^{supp}|$ is the number of unique edges.

Thus, the total computational cost of our two-stage aggregation is $O(|\mathcal{E}|d + |\mathcal{E}^{supp}|d)$, which simplifies to $O(|\mathcal{E}|d)$ since $|\mathcal{E}^{supp}| \leq |\mathcal{E}|$. This shows that the asymptotic complexity of our method is comparable to that of the single-stage aggregation methods, ensuring scalability to large multigraphs.

## D.2   MEMORY OVERHEAD

In terms of memory, we assume that the multigraph is attributed, with feature vectors associated with each edge. The memory required to store the edge embeddings is $O(|\mathcal{E}|d)$ in the standard case.

For the two-stage aggregation, additional memory is needed to store a tensor of size $|\mathcal{E}|$ that indexes parallel edges in the multiset. This tensor is computed during preprocessing and reused across batches, thus avoiding redundant calculations. Furthermore, a tensor of size $|\mathcal{E}^{supp}| \times d$ is created dynamically during the forward pass to store the features of the artificial nodes corresponding to distinct neighbors.

As a result, the overall memory overhead per batch is $O(|\mathcal{E}|d + |\mathcal{E}| + |\mathcal{E}^{supp}|d) = O(|\mathcal{E}|d)$, which scales linearly with the number of sampled edges. Importantly, this does not asymptotically alter the overall memory complexity, meaning that the model remains efficient even as the size of the input graph increases.

## D.3 Optimizations for Large-Scale Data

While the computational and memory overheads of our method are manageable in practice, further optimizations could be explored for very large-scale data. Techniques such as kernel fusion on GPUs and the design of high-speed multi-stage aggregation kernels with low memory footprints could significantly mitigate these overheads. However, such optimizations are beyond the scope of this work and are left as promising directions for future research.

## E Ablation Experiment

In this section, we perform additional experiments on the baseline method, Multi-GNN, to investigate the effect of randomly assigning port-ids. The results, presented in Table 8, were obtained using the same set of hyperparameters as those used for MEGA-GIN and MEGA-PNA, applied to Multi-GIN and Multi-PNA, respectively.

| Ablation | Small_HI | Small_LI | ETH-Kaggle | Medium_HI | Medium_LI |
|---|---|---|---|---|---|
| Multi-GIN | $62.65 \pm 1.73$ | $32.21 \pm 0.99$ | $51.34 \pm 3.92$ | $67.72 \pm 0.93$ | $31.24 \pm 2.12$ |
| Multi-GIN (permuted) | $61.36 \pm 4.17$ | $29.00 \pm 3.49$ | $49.13 \pm 3.72$ | $67.58 \pm 1.96$ | $30.19 \pm 1.49$ |
| Multi-PNA | $67.35 \pm 2.89$ | $35.39 \pm 3.93$ | $64.61 \pm 1.40$ | $71.93 \pm 1.54$ | $43.81 \pm 0.51$ |
| Multi-PNA (permuted) | $66.58 \pm 1.86$ | $33.05 \pm 3.84$ | $65.01 \pm 0.97$ | $73.41 \pm 1.38$ | $42.61 \pm 1.21$ |

Table 8: Multi-GNN ablations: The port-id assignment in Multi-GNN follows a strict total ordering among edges. In the permuted ablations, port-id assignment is randomized to evaluate the impact of edge ordering on model performance.

Randomly permuting port-id assignments leads to a slight performance drop for both models, with Multi-GIN experiencing a more significant decline compared to Multi-PNA. Notably, Multi-PNA demonstrates greater robustness to permuted port-ids. Interestingly, on the ETH-Kaggle dataset, permuting the port-ids results in a small performance improvement for the PNA-based model, suggesting that the permutation equivariance property is less critical for this particular dataset.

