# OpenReview forum: "Multigraph Message Passing with Bi-Directional Multi-Edge Aggregations"
_ICLR.cc/2025/Conference — Submitted to ICLR 2025_

### Official Review · Reviewer_Euww · 2024-10-29

**Soundness:** 2
**Presentation:** 2
**Contribution:** 2
**Rating:** 3
**Confidence:** 4

**Summary:**

This paper proposes a novel message-passing procedure on multigraphs. By first aggregating all edge states between two nodes, and using those as messages for MPNNs, a permutation equivariant method is constructed. The effectiveness of the proposed method is shown on several edge classification and one node classification task.

**Strengths:**

I like the proposed method. It is intuitive, simple and effective. The paper is nicely written and mostly easily to follow and sufficiently detailed described.

**Weaknesses:**

W1 Theorem 3.1: All MPNNs are permutation equivariant, as they do not use indices directly. I do not understand the need to specifically point this out.

W2 Corollary 3.1: This statement is false the way it is stated. First, you need f, g_v, and g_e to be injective as well. Lemma 5 of Xu et al. also only holds for functions over countable multisets, which is ignored in this statement. In fact, if you follow Lemma 5 of Xu et al., there needs to be an additional function that is applied before EdgeAgg for possible injectivity.

W3 Theorem 3.2: It should be clearly defined what the authors mean with universality. If a consistent ordering of the edges is given, permutation equivariance is lost. I find the claim that "our method is universal and capable of detecting any directed subgraph pattern in multigraphs" to be strongly misleading. How to construct a consistent ordering is not discussed and from my understanding not used for the experiments. Therefore, the point of this theoretical statement is unclear to me.

W4: Eq. 4 computes h^(l) but Eqs. 5,6 use h^(l-1). Is this intended?

W5: The introduced artificial nodes are not used in the proposed method. States are only computed for edges and nodes, but not for the artificial ones. Discarding the artificial nodes would make the framework much cleaner.

W6 Experiments:
* The baseline results seem to be reused from previous methods. It is unclear whether the same dataset splits were used, the same number of parameters, and the same hyperparameter search space.
* There are no ablation studies. For example, to better understand this method it would be nice to see how results would change if there was a single edge state between two nodes instead of having multiple edge states.

**Questions:**

* l. 376: What does GIN aggregation mean? GIN uses sum aggregation + an MLP.
* MEGA-GNN seems to require a single vector of edge features, independently of the number of edges between two nodes. From my understanding all edge between two nodes would be equal if there are no initial edge features for each edge. This seems to a large constraint and it is not discussed in the experiments. Are these given for all datasets? Are baseline methods utilizing those as well?
* see weaknesses

---

> ### Author Response · Authors · 2024-11-18
> **Author Response 1/2**
>
> We thank the reviewer for the insightful feedback and points. We are very pleased that you find our paper intuitive and effective. We address your concers below.
>
> > **W1**: Theorem 3.1: All MPNNs are permutation equivariant, as they do not use indices directly. I do not understand the need to specifically point this out.
>
> - In Proposition 2.1, we demonstrate that the baseline method, Multi-GNN, is not permutation equivariant when there is no natural or contextually driven edge ordering (e.g., timestamps). In such cases, permuting the edges alters the multigraph port IDs assigned by Multi-GNN, which breaks the permutation equivariance property at the model level. While the message-passing scheme itself remains permutation equivariant, the overall Multi-GNN architecture loses this property due to its port-ID assignment mechanism. In contrast, our model preserves permutation equivariance regardless of the ordering of edges or nodes. This distinction is critical for ensuring the robustness of graph-based models in contexts where edge ordering is not naturally defined.
>
> > **W2**: Corollary 3.1: This statement is false the way it is stated. First, you need f, g_v, and g_e to be injective as well. Lemma 5 of Xu et al. also only holds for functions over countable multisets, which is ignored in this statement. In fact, if you follow Lemma 5 of Xu et al., there needs to be an additional function that is applied before EdgeAgg for possible injectivity.
>
> - We would like to thank the reviewer for pointing out this issue. We agree that for the overall message passing scheme to be injective, f, g, and g_e functions must also be injective. Furthermore, proving injectivity over multi-sets would require additional considerations. We would like to clarify that injectivity is not a foundational argument of our paper and is not used to derive or support any other theoretical results, such as equivalence to the Weisfeiler-Lehman (WL) test or adaptations for multigraphs. Given that this theorem is not central to the core contributions of the paper and does not connect to other results, we will remove it along with its informal proof to avoid confusion.
>
> > **W3** Theorem 3.2: It should be clearly defined what the authors mean with universality. If a consistent ordering of the edges is given, permutation equivariance is lost. I find the claim that "our method is universal and capable of detecting any directed subgraph pattern in multigraphs" to be strongly misleading. How to construct a consistent ordering is not discussed and from my understanding not used for the experiments. Therefore, the point of this theoretical statement is unclear to me.
>
> - The definition of universality we consider is based on [1], where universal MPNNs are defined as those can approximate every function defined on graphs of any fixed order. According to [1], MPNNs are proven to be universal if they satisfy the following conditions: they have enough layers with sufficient expressiveness and width, and nodes can uniquely distinguish each other. In our setup, we also assume there are enough layers with sufficient expressiveness and width. Theorem 3.2 shows that if a consistent ordering of edges exists, MEGA-GNN can leverage this ordering to assign unique node IDs, making the MEGA-GNN universal in this sense.
> - Regarding the concern about permutation equivariance: the model does not explicitly rely on edge/node indices or their ordering. Instead, edge features like timestamps naturally encode a consistent ordering in some datasets (e.g., transaction data). This ordering is implicit and does not affect the permutation equivariance of the model, as the MEGA-GNN operates independently of the specific indexing of the edges/nodes. For clarification, by “consistent ordering of the edges,” we refer to a natural sequence such as timestamps, rather than arbitrary edge indices. This distinction ensures that permutation equivariance and universality are orthogonal concepts and can coexist within the model.
> -  if the reviewer finds it helpful, we can add further clarifications to the paper.
>
> > **W4**: Eq. 4 computes h^(l) but Eqs. 5,6 use h^(l-1). Is this intended?
> - Thank you for pointing it out. No, this is not intended. There is a typo in the equations, it should be h^(l-1) in Eq. 4 and Eq. 8., they are corrected in the current version.
>
> [1] What graph neural networks cannot learn: depth vs width. ICLR'20

---

> ### Author Response · Authors · 2024-11-18
> **Author Response 2/2**
>
> > **W5**: The introduced artificial nodes are not used in the proposed method. States are only computed for edges and nodes, but not for the artificial ones. Discarding the artificial nodes would make the framework much cleaner.
> - The concept of artificial nodes, created dynamically, provides an intermediate representation for aggregated parallel edges. Their role is explained in detail in lines 216–227 and visually illustrated in Figure 4. We encourage the reviewer to revisit Figure 4, as it complements the textual explanation and demonstrates how artificial nodes contribute to the model's formulation.
>
> > **W6.1** Experiments: The baseline results seem to be reused from previous methods. It is unclear whether the same dataset splits were used, the same number of parameters, and the same hyperparameter search space.
>
> - We used temporal splits for the datasets, consistent with the baseline method Multi-GNN, following the dataset splitting strategy from its open-sourced implementation. Specifically, we applied splits of 64/19/17 (AML-Small) and 61/17/22 (AML-Medium) on the AML datasets, and 65/15/20 on the ETH dataset, based on edges (transactions). This ensures that our experiments are directly comparable to the baseline. Line 338 is updated with this information. We will add further clarification into the implementation part of the paper (l. 337.).
>
> - The results in Table 2 of the paper are adopted from the previous method, Multi-GNN. The hyperparameters used are detailed in Appendix B.1. For MEGA-PNA, we kept the same hyperparameters as in Multi-PNA. However, for the MEGA-GIN, we made minor adjustments compared to Multi-GIN, specifically reducing the learning rate and modifying the hidden dimension (as shown in Table 1 below). These changes are unlikely to have a significant impact on the results, but we are open to repeating the experiments with the hyperparameters from MEGA-GIN for Multi-GIN to ensure clarity.
>
> **Table 1: Comparison of Hyperparameters for Multi-GIN and MEGA-GIN **
>
> | Hyperparameter | Multi-GIN | MEGA-GIN |
> |----------------|----------|---------|
> | Learning Rate (lr) | 0.006    | 0.003   |
> | Hidden Dimension (n_hidden) | 66       | 64      |
>
> - In case of Table 3, we did not reuse the results of the Multi-GNN paper, as the ETH dataset used in our experiments is different from the one used in the Multi-GNN paper. To make our experiments reproducable, we used the open-source ETH dataset available on Kaggle without any changes. In contrast, the authors of Multi-GNN appear to have enhanced that dataset with additional features and transactions collected from the Ethereum block chain. Since this enhanced dataset is not publicly available, we opted to use the original dataset from Kaggle and repeated all the experiments.
>
> > **W6.2** Experiments: There are no ablation studies. For example, to better understand this method it would be nice to see how results would change if there was a single edge state between two nodes instead of having multiple edge states.
>
> -  We are considering edge-attributed multigraphs, where multiple parallel edges exist between two nodes, each associated with distinct edge features (states). The multi-edge aggregation process is meaningful only if initial edge features are present.
>
> - For the ablation studies, we will include the following experiments:
>     - The effect of permuting the validation and test set edges for MEGA-GNN and Multi-GNN on the AML Small datasets. In this experiment, Multi-GNN will lose its permutation invariance property, whereas MEGA-GNN will still retain it.
>     - MEGA-GNN with/without reverse-message passing and MEGA-GNN with/without Ego-IDs on the AML Small datasets.
> - If you have any other suggestions we are happy to hear them.

---

> ### Author Response · Authors · 2024-11-18
> **Author Response to Questions**
>
> > **Q1**: What does GIN aggregation mean? GIN uses sum aggregation + an MLP.
>
> - What we mean by "GIN aggregation" is a sum aggregation + an MLP, where we first sum the parallel edges and then pass the result through an MLP. The details of this process are provided in Appendix B.2 of our submission. We can also include a reference to Appendix B.2 in line 376 for clarity.
>
> > **Q2**: MEGA-GNN seems to require a single vector of edge features, independently of the number of edges between two nodes. From my understanding all edge between two nodes would be equal if there are no initial edge features for each edge. This seems to a large constraint and it is not discussed in the experiments. Are these given for all datasets? Are baseline methods utilizing those as well?
>
> - MEGA-GNN is designed to work with edge-attributed multigraphs, where each edge is associated with a feature vector. Parallel edges between two nodes are distinct, with each edge having a different feature vector that corresponds to it, as they represent different interactions. For example, sender A might send money to receiver B multiple times, with varying amounts.
>
> - The multi-edge aggregation process is meaningful only if initial edge features are present, as its primary goal is to extract higher-order patterns from the features of parallel edges and enhance message passing within the multigraph context. For the AML datasets, the edge features include: ['Timestamp', 'Amount Received', 'Received Currency', 'Payment Format']. For the ETH dataset, the edge features are: ['Amount', 'Timestamp']. All baseline methods also utilize the same set of features.
>
> - To resolve any potential misunderstanding, we will revise line 190 to explicitly state that each parallel edge has distinct features, ensuring this aspect is clearly explained.

---

> > ### Comment · Reviewer_Euww · 2024-11-20
> >
> > Thank you for the detailed response. While looking deeper into the work and the reviews, I have a small follow-up question that I would like to have clarified before I can further assess this work.
> >
> > Consider ADAMM, but instead of applying DeepSet a single time before message-passing, it is applied in each layer and separately for each edge direction. Outputs are used for your Eq. 5. Edge states are also calculated individually per edge in each layer, based on your Eq. 6. Would this be equivalent to MEGA-GNN? If not please describe the differences. To me, it seems equivalent and might be a cleaner formulation for the proposed method.

---

> ### Author Response · Authors · 2024-11-20
> **Author Response**
>
> We thank the reviewer for their interest in further assessing our work. There are three main differences between our work and ADAMM:
> 1. Unlike ADAMM, which applies a form of pre-embedding aggregation that collapses parallel edges into single edges before message passing, we implement multi-stage aggregation directly within the message-passing layers. This enables performing multi-stage aggregation repeatedly across several message passing layers while preserving the underlying graph topology.
> 2. We perform multi-stage aggregation in a bi-directional manner for directed multigraphs by introducing a separate artificial node to aggregate the parallel edges in both directions.
> 3. While ADAMM relies only on DeepSet, our approach leverages state-of-the-art aggregation functions such as PNA and GenAgg, which offer more general and learnable aggregation mechanisms.
>
> As a result, our methods are not only more accurate than ADAMM, but can also perform additional fundamental tasks such as edge classification or edge embedding generation, which are not supported by the solution presented in ADAMM.

---

> ### Comment · Reviewer_Euww · 2024-11-25
>
> I again thank the reviewer for their responses. After thoroughly looking into all reviews, the author's responses, and the paper, I have decided to maintain my score. While I do think that the method generally makes sense, my main concerns remain:
>
> 1) To me, the paper seems unnecessarily complicated for proposing to apply a permutation invariant aggregation on the edge representations.
> 2) All three theoretical statements are not insightful to me.

---

> ### Author Response · Authors · 2024-11-25
> **Oversimplification**
>
> We do not simply apply a permutation invariant aggregation on the edge representations. That is what ADAMM had already done. As indicated earlier, we outperform ADAMM significantly and introduce additional capabilities that ADAMM does not have.
>
> We agree that the motivation of our theoretical analysis and our insights can be better expressed, which we will address in the final manuscript we are going to submit soon. We will also include additional ablation studies to further clarify our improvements.

---

### Official Review · Reviewer_xLVR · 2024-11-02

**Soundness:** 3
**Presentation:** 3
**Contribution:** 3
**Rating:** 6
**Confidence:** 3

**Summary:**

The authors introduce MEGA-GNN, a message-passing framework for multigraphs—graphs where multiple parallel edges can exist between the same pair of nodes. Traditional GNNs are not well-suited for multigraphs due to their single-stage node-level aggregation.

MEGA-GNN addresses this limitation by implementing a two-stage aggregation process: first, it aggregates parallel edges between the same nodes, and then it performs node-level aggregation on the aggregated messages from distinct neighbors. The authors give theoretical proofs that MEGA-GNN supports permutation equivariance, injectivity, and universality under specifi conditions.

Experimental results on synthetic and financial transaction datasets demonstrate that MEGA-GNN either outperforms or matches the accuracy of state-of-the-art models.

**Strengths:**

1. MEGA-GNN gives a solution to a gap of GNNs' effecitveness, by focusing on multigraphs rather than simple graphs, where multiple edges between nodes could be useful for many applications, such as financial fraud detection (as benchmarked in the paper).

2. The authors theoretically validate MEGA-GNN’s permutation equivariance, injectivity, and universality properties.

3. MEGA-GNN achieves state-of-the-art or comparable performance across various (synthetic and real-world0 datasets, particularly in edge and node classification tasks.

**Weaknesses:**

1. The proof for the universality property assumes that there is an ordering over the edges of the multigraph. This is not always the case for real-world setups, giving a small question-mark on what happens when the edge ordering is not consistent.

2. Although I find the utilization of financial transaction datasets quite interesting, I think it's not very diverse. I'd be very interested in seeing whether such a multigraph approach can be useful in other domains (e.g. knowledge graphs could potentially be of relevance due to the variety and number of different relations that can occur between identical pairs of nodes. For example, biomedical knwoeldge graphs could a potential use).

3. The method requires the addition of artificial nodes, and the two-stage aggregation. This hints a computational overhead that is not discussed in the paper, and how it'd affect message passing in large/dense graphs.

**Questions:**

1. How does MEGA-GNN scale with larger, more complex multigraphs, and what are the computational costs with the two-stage aggregation process?

2. Is it possible to relax the assumption of consistent edge ordering, and if so, how would that impact the theoretic properties of MEGA-GNN?

---

> ### Author Response · Authors · 2024-11-21
> **Author Response**
>
> We thank the reviewer for their constructive feedbacks. We address your concers below.
>
> **W1:** The universality proof is a theoretical demonstration of our model's capability under certain assumptions. Specifically, it shows that MEGA-GNN is universal if a natural edge ordering (e.g., timestamps) exists. However, in practice, MEGA-GNN does not rely on the explicit ordering or indexing of edges/nodes to produce results. Instead, it operates independently of any specific edge ordering. In real-world setups where a natural ordering is not present, the theoretical universality property no longer holds.
>
> **W2:** Our approach is orthogonal to, and can complement, relational GNNs applied to knowledge graphs, especially when edge attributes and multiple edge types exist between the same nodes. This also includes scenarios where different relations between the same node pairs can be aggregated. However, the availability of well-established, labeled edge-attributed multigraph benchmark datasets is limited, particularly for edge and node classification tasks relevant to our work. Moreover, our model is especially effective when edges have distinct features. In the knowledge graphs we examined from OGB and TGB, the majority of edge types lack explicit feature vectors, with only a small subset having such attributes, which restricts the applicability of our approach to these benchmarks.
>
> **W3:** We kindly refer the reviewer to our response to Reviewer 74YN in W1, where we discuss the scalability and memory overhead of our method in detail.
>
> ---
>
> **Q1:** We kindly refer the reviewer to our response to Reviewer 74YN in W1, where we discuss the scalability and memory overhead of our method in detail.
>
> > If the reviewer finds these discussions helpful, we would be happy to include them in the paper. Additionally, we can provide inference throughput and memory consumption measurements and comparisons.
>
> **Q2:** If the assumption of consistent edge ordering is relaxed, the model would lose the universality property, but still retain permutation equivariance. However, in practice, MEGA-GNN does not rely on the explicit ordering or indexing of edges/nodes to produce results. Instead, it operates independently of any specific edge ordering. We included the universality proof in our submission to show that MEGA-GNN would be as powerful as Multi-GNN under the consistent edge ordering assumption.

---

### Official Review · Reviewer_74YN · 2024-11-04

**Soundness:** 3
**Presentation:** 4
**Contribution:** 3
**Rating:** 6
**Confidence:** 3

**Summary:**

The paper introduces MEGA-GNN, a novel message-passing framework tailored for multigraphs—graphs that include multiple parallel edges between node pairs. Unlike standard GNNs that aggregate all edges at once, MEGA-GNN introduces a two-stage aggregation strategy: Parallel Edge Aggregation and Node-Level Aggregation. The authors demonstrate the permutation equivariance and invariance of the proposed model and show that it is universal when the edges are consistently ordered.

**Strengths:**

- The paper is clearly written and easy to follow with various figures demonstrating the ideas in the paper.
- Novel Aggregation Mechanism: The two-stage approach addresses the limitation of traditional GNNs, enhancing expressivity by first aggregating parallel edges and then aggregating at the node level.
- The paper provides proofs for permutation equivariance, injectivity, and universality.
- Experimental Evaluation: The proposed method shows promising improvements in the included datasets.
- The code is included in the supplementary material.

**Weaknesses:**

1- Scalability: Although multigraphs are well-suited to some applications, real-world graphs can be vast in scale. The two-stage aggregation with artificial nodes might pose computational challenges and memory overhead for large, densely connected multigraphs. Some discussion on scalability in practical settings or optimizations for large-scale data would be appreciated.

2- The paper shows that under a consistent ordering of edges the model is universal. However, for many real work scenarios, this is not always feasible, especially in dynamic setting. Have authors considered dynamically evolving multigraph setting?

3- The experiments are limited to financial datasets and lack diversity in application areas, which might constrain the broader applicability of MEGA-GNN. I am not familiar with the multi-graph learning literature but are there other domains you can explore?

**Questions:**

See weaknesses.

---

> ### Author Response · Authors · 2024-11-21
> **Author Response**
>
> We thank the reviewer for their constructive feedbacks. We address your concers below.
>
> **W1:**
>
> **Discussion on Scalability:**
> - Let $\mathcal{V}$ denote the node set, $\mathcal{E}$ denote the multiset of edges in the sampled batch, and $\hat{\mathcal{E}}$ denote the set of edges obtained by removing duplicated entries from $\mathcal{E}$, $d$ be the dimensionality of edge embeddings. We compare the asymptotic computational complexity for the aggregation functions sum, std, max, and min, which are the default methods used in the PNA framework.
>     - The computational cost of performing single-stage aggregation $O(|\mathcal{E}|d)$ for functions (sum, std, min, max).
>     - In our two-stage aggregation:
>         1.  Parallel Edge Aggregation: Aggregating parallel edges incurs a cost of $O(|\mathcal{E}|d)$ for functions (sum, std, min, max).
>         2.  Distinct Neighbor Aggregation: Aggregating over distinct neighbors involves a cost of $O(|\hat{\mathcal{E}}|d)$ for functions (sum, std, min, max).
>     - Thus the total cost is $O(|\mathcal{E}|d + |\hat{\mathcal{E}}|d) = O(|\mathcal{E}|d)$ since $|\hat{\mathcal{E}}| \leq |\mathcal{E}|$. This demonstrates that our method has the same asymptotic complexity as single stage sum, std, min and max aggregations, ensuring scalability to large multigraphs.
>
> - Memory Overhead:
>     - We assume the multigraph is attributed, with feature vectors associated with each edge. In the standard case, storing the edge embeddings requires $O(|\mathcal{E}|d)$ memory.
>     - In the two stage aggregation we additionally store a tensor of size $|\mathcal{E}|$ to index parallel edges in the edge multiset. This tensor is computed during preprocessing and reused in each batch, eliminating the need for redundant computations. During the forward pass, we dynamically create a tensor of shape $|\hat{\mathcal{E}}| \times d$ to store the features of artificial nodes.
>     - Therefore, the overall memory overhead per batch is $O(|\mathcal{E}|d + |\mathcal{E}|+|\hat{\mathcal{E}}|d)=O(|\mathcal{E}|d)$, which scales linearly with the number of sampled edges, this does not asymptotically change the overall memory complexity of the stored tensors.
>
> **Discussion on Optimizations for Large-Scale Data**
> - We believe that these overheads could be mitigated by using kernel fusion techniques on GPUs and that designing high-speed multi-stage aggregation kernels with low memory footprints would be a promising research direction. However, such optimizations are outside the scope of this work.
>
> If the reviewer finds these discussions helpful, we would be happy to include them in the paper. Additionally, we can provide inference throughput and memory consumption measurements and comparisons.
>
> **W2:** A consistent ordering of the edges is possible also in a dynamic setting. One can think about timestamps assigned to transactions, where future transactions will have higher timestamp values than the past ones. These timestamps can also be unique if the precision of the measurements is high enough.
>
> - As noted in the paper (limitations, line 467), we did not specifically address dynamically evolving multigraphs. Instead, we temporally split the entire transaction history for all datasets in our experiments into train, validation, and test graphs. The data split is defined by two timestamps t1 and t2. This corresponds to considering the financial transaction graph as a dynamic graph and taking three snapshots at times t1, t2, and t3 = tmax. This approach represents an inductive setting, where the test graph contains unseen transactions and/or accounts.
> -  In principle, the MEGA-GNN model itself could support dynamic settings. However, adapting our implementation to a truly streaming setting would require certain adjustments, e.g., the neighbourhood sampling schemes we use  would have to enforce causality or time window constraints. We are open to elaborating on this limitation if needed.
>
> **W3:** There could be interesting applications of multigraphs in cybersecurity (e.g., several different  connections between the same two network ports), social network analysis (several different interactions between the same two persons), and biological network analysis (different types of interactions between the same two proteins). However, there is a limited availability of well-established and labeled edge-attributed multigraph benchmark datasets, especially for the edge and node classification tasks we covered. In our comparisons with Multi-GNN, we used the same financial transaction datasets they used: synthetic (AML) datasets recently introduced at NeurIPS'23, and one real-world dataset from Ethereum transactions. We also considered using OGB and TGB benchmarks, but these datasets do not provide edge-attributed multigraphs with node or edge labels.

---

### Official Review · Reviewer_Q2cP · 2024-11-08

**Soundness:** 3
**Presentation:** 3
**Contribution:** 2
**Rating:** 3
**Confidence:** 5

**Summary:**

This paper studies neural architecture for learning on multigraphs. Existing methods either reduce to simple graphs or break properties such as permutation equivariance. A two-stage message-passing framework is proposed by introducing artificial nodes for parallel edges that preserves several desirable properties. The proposed MEGA-GNN is evaluated on synthetic and real-world financial transaction datasets and shows better or comparable performance compared to SoTA methods.

**Strengths:**

- The use of two-stage aggregation and artificial nodes can effectively address the limitations of existing GNN models to capture information across parallel edges while preserving certain properties.

- The MEGA-GNN framework shows good flexibility and performance in applications on financial transaction datasets.

- The authors offer in-depth analysis and jusfiication for the proposed framework regarding properties of permutation equivariance, injectivity and universality.

**Weaknesses:**

- The techniques used in MEGA-GNN such as bi-directional message passing and multi-stage aggregation are already well established, and the technical challenges for multi-graph have also been largely addressed by hypergraph learning research, which limits the overall novelty.

- The model is only evaluated on financial datasets, which raise questions about wheter multi-graph learning is applicabile for broad scenarios.

- Some choices, such as specific aggregation functions and the role of artificial nodes, lack detailed justification.

Feng, Yifan, et al. "Hypergraph neural networks." AAAI’19
Gao, Yue, et al. "Hypergraph learning: Methods and practices." TPAMI’2020.

**Questions:**

- Q1 In practice, is permutation equivariance indeed needed for multi-graph applications? Based on results from Tables 2 amd 3, Multi-GNN without this property is a very strong baseline, especially for the node classification task.

- Q2 What aggregation functions (EdgeAgg in Eqs 4, 8, AGG in Eqs 5,9) are used for the results reported in Tables 2 and 3?

- Q3 Could the authors include ablation studies to highlight the contribution of Ego-IDs? Comparsion with other baselines without node labelling seems unfair.

You, Jiaxuan, et al. "Identity-aware graph neural networks." AAAI'21

---

> ### Author Response · Authors · 2024-11-13
>
> We first would like to address a potential misunderstanding by pointing out the differences between Hypergraph GNNs and our Multi-Graph GNNs:
> - Hypergraphs are not multigraphs and multigraphs are not hypergraphs.
> - A hyperedge is a collection of nodes whereas a multi-edge is a collection of edges.
> - Hypergraph GNNs first aggregate feature vectors of the nodes in each hyperedge to compute latent representations of the hyperedges and then aggreate the latent representations of the hyperedges to compute latent representations of the nodes.
> - Our multigraph GNNs (MegaGNNs) first aggregate the feature vectors of the edges between the same pair of nodes to compute late representations of multi-edges and then aggregate latent represantions of the multi-edges to compute the latent representations of the nodes.
> - Hypergraphs are generalizations of graphs and not multigraphs. Hypergraphs with repeated hyper edges (multi-hypergraphs) would be generalizations of multi-graphs. However, the papers the reviewer listed do not cover multi-hypergraph GNNs.
> - It is possible to combine our multigraph GNNs with hypergraph GNNs to construct multi-hypergraph GNNs. However, that would - require three stages of aggregations: 1) hyperedge aggregations, 2) multi-hyperedge aggregations, and 3) node aggregations. A mechanism of this form is not available in the papers cited by the reviewer or in other papers we have seen.
> - We hope that we have sufficiently expressed the differences between hypergraph GNNs and our multi-graph GNNs and clarified that these two approaches are orthogonal. If the reviewer finds it convenient, we will include some of these explanations also in the paper.

---

> > ### Comment · Reviewer_Q2cP · 2024-11-13
> >
> > Thank you for the detailed response on the differences between multi-graphs and hypergraphs, especially regarding the different structural features of hyperedges vs. multi-edges and the specific aggregation hierarchies each requires.
> >
> > However, my main point remains that the core techniques employed in MEGA-GNN, such as bi-directional message passing and multi-stage aggregation, are already well-established in this line of research. The fundamental approach of multi-level aggregation for non-simple graphs (e.g., hypergraphs and heterogeneous graphs) has been extensively explored. In this regard, I find it difficult to view MEGA-GNN’s approach as conceptually novel, as it builds on established methodologies without clearly demonstrating unique functional advantages in handling multi-edges. It would be great to explicitly highlight any specific technical challenges that are unique in multi-graph contexts and show how the design of MEGA-GNN addresses these challenges differently than existing solutions. Further empirical studies, such as ablation, could also help clarify the unique contributions of this work, especially if they go beyond adaptations of aggregation mechanisms.

---

> ### Author Response · Authors · 2024-11-14
> **Official Comment by Authors**
>
> We appreciate the reviewer’s comments and would like to clarify the novelty of our approach regarding bi-directional message passing and multi-stage aggregation within a multigraph context.
> - The techniques developed for hypergraphs and heterogeneous graphs are not directly applicable to multigraphs. As we have already covered the differences between multigraph GNNs and hypergraph GNNs in our previous response, we will cover only heterogeneous graphs here. Relational GNNs proposed for heterogeneous graphs work very differently from our multigraph GNNs. Relational GNNs require definition of explicit relation (edge) types and use different transformation functions for different relation types. Our approach directly works on edge attributes, does not require definition of any edge types, and does not differentiate between different relations. Our approach is orthogonal and can be combined with relational GNNs on heterogeneous graphs with edge attributes when multiple edges of the same type (with different edge attributes) exist between the same two graph nodes.
> - In Section 2 of our paper, we have described the challenges associated with supporting multi-edges, the prior work specifically designed for this purpose (Multi-GNN and ADAMM), and where they fall short. Our main contribution is the introduction of artificial nodes, which enable multi-edge aggregations in both forward and reverse directions. This mechanism is nontrivial and new, and was not part of any previous solutions. Notably, unlike the prior solutions, our approach preserves both the permutation equivariance and the multigraph topology. Furthermore, it is efficiently integrated into multigraph message passing. Our MEGA-GNNs lead to consistent improvements, in the range of 9-13%, in comparison to Multi-GNNs as shown in Table 2. We also achieve around 20% improvement with respect to ADAMM as shown in Table 3. We believe these improvements clearly show the significance of our contribution.
> - We would be glad if the reviewer could be more specific about the type of ablation studies requested. However, detailed ablation studies evaluating the performance impact of using Ego-IDs, reverse message passing, and multigraph port numbering in combination with different baseline GNNs are readily available in the Multi-GNN paper. In this paper, we have tried to build on the lessons learned from Multi-GNN and ADAMM, and demonstrated our improvements with respect to these two methods.

---

> > ### Author Response · Authors · 2024-11-14
> > **Author Response 1/2**
> >
> > We appreciate the reviewer’s effort in reviewing our paper, thank them for their comments. We have carefully considered each point and addressed them individually to ensure clarity.
> >
> > ### W1
> > - Two-Stage Aggregation for Multigraphs: To the best of our knowledge, MEGA-GNN is the first MPNN to apply a two-stage aggregation process for message passing on multigraphs. MEGA-GNN is designed to handle the unique complexity of multiple connections between the same pair of nodes -which is not fully addressed by prior research.
> > - Bi-directional Message Passing with Multi-Edge Aggregations: MEGA-GNN’s bi-directional message passing approach involves non-trivial and novel mechanisms. While earlier approaches, such as Multi-GNN, also utilize bi-directional message passing, our method differentiates itself by incorporating two artificial nodes for each directed multi-edge: one for aggregating the forward messages and one for aggregating the backward messages. This inclusion goes beyond merely using the reverse edges; it directly integrates these reversed edges into our two-stage message aggregation mechanism.
> > - We hope this response clarifies the originality of MEGA-GNN’s bi-directional message passing and its two-stage aggregation process.
> >
> > ### W2
> > - We acknowledge the reviewer’s concern about the general applicability of multigraph learning. There is a limited availability of well-established and labeled edge-attributed multigraph benchmark datasets for edge and node classification tasks. To provide fair comparisons with baseline methods such as Multi-GNN, we used the same financial transaction datasets they used: synthetic (AML) datasets recently introduced at NeurIPS'23, and one real-world dataset from Ethereum transactions, as real-world financial data is often restricted due to privacy concerns. We also considered using OGB and TGB benchmarks, but these datasets do not provide edge-attributed multigraphs with node or edge labels. If the reviewer could suggest some edge-attributed multigraph datasets with edge or node labels, we would be happy to evaluate our solutions on them.
> >
> > ### W3
> > - We chose GIN aggregation as a strong baseline widely used in graph learning across various tasks. Additionally, we adopted PNA aggregation because the authors of Multi-GNN paper demonstrated its superior performance. To further enhance our model, we also incorporated the state-of-the-art learnable aggregation function, Generalized Aggregator (GenAgg). The choice of the aggregator depends on the task as aggregation is a lossy operation. GenAgg, a generalized operator, parametrizes a function space that includes all standard aggregators, and allows to minimize information loss effectively. By doing so, we believe we have covered the most important aggregation functions, yet if the reviewer has any other suggestions, we would be happy to explore them.
> > - The artificial nodes serve as intermediate feature representations for the aggregated parallel edges in our model. Their role is detailed in lines 210-227 of the paper.

---

> ### Author Response · Authors · 2024-11-14
> **Author Response 2/2**
>
> ### Q1
> - In Table 2, MEGA-GNN significantly outperforms all the baselines. Specifically, our model achieves an average increase of 9.25% in minority-class F1 score on the HI datasets and a 13.31% improvement on the LI datasets over the best-performing baseline, Multi-GNN.
> - While Multi-GNN is permutation equivariant only if there is a natural ordering of the edges, e.g., defined by timestamps, MEGA-GNN satisfies the permutation equivariance property without requiring such an ordering. In our experiments, the financial transaction datasets include timestamps, providing a natural edge ordering. If the reviewer finds it useful, we could perform an ablation study by removing the timestamps, which would cause Multi-GNN to lose permutation equivariance, while MEGA-GNN would still retain it. We would be happy to repeat the experiments on the AML Small datasets for edge classification and on the ETH dataset for node classification under this condition.
> - In Table 3, MEGA-GNN performs on par with the Multi-GNN baseline and shows a 20% improvement over ADAMM. This specific dataset has an average node degree of 4.5. When the node degree is 4 or smaller, a single-level PNA aggregator (which combines 4 standard aggregators: sum, min, max, and std) and the following MLP can effectively reconstruct the multiset of neighbors without any loss, as shown in the PNA [1] paper (Appendix A). Thus, for this dataset, the single-level PNA aggregation is as effective as the two-stage PNA aggregation both in theory and in practice. If the reviewer finds it useful, we can also include a theoretical analysis to further explain the impact of dataset characteristics on performance and include these insights in the paper.
> ### Q2
> - The multi-edge aggregation functions used in our experiments are described in lines 375-377 of our paper. For more detailed information on the specific EdgeAgg functions, please refer to Appendix B.2.
> ### Q3
> - The impact of Ego-IDs has already been studied in the Multi-GNN paper, so we did not repeat those ablation studies in our work. However, if the reviewer finds it helpful, we can either include these studies in the paper or mention which baselines also adopt Ego-IDs to ensure clarity and fairness in the comparison
>
> [1] Principal Neighbourhood Aggregation for Graph Nets, NeurIPS 2020

---

> > ### Comment · Reviewer_Q2cP · 2024-11-27
> >
> > Thank the author for the detailed response. I appreciate the effort in clarifying MEGA-GNN and existing approaches for hypergraphs and heterogeneous graphs. However, I still have some reservations regarding the novelty and the articulation of contributions specific to multigraphs. Thus, I intend to maintain my original rating.
> >
> > I would like to outline some key aspects where additional clarity and evidence could enhance the manuscript:
> >
> > **Applicability of Existing Techniques:**
> > While you emphasize the differences between techniques used for hypergraphs, heterogeneous graphs, and multigraphs, I remain unconvinced that the adaptations made in MEGA-GNN significantly diverge (or "orthogonal") from established methodologies. For instance, bi-directional message passing is described as analogous to relational GNN approaches (line 258). I recommend a deeper exploration into the unique challenges that multigraphs present, which are not addressed by existing solutions, to highlight the novelty part better.
> >
> > **Two-Stage Message Passing:**
> > The transition to two-stage message passing seems to parallel existing strategies in HGNNs: "node group->hyperedge group->node group over the star-expanded graph" vs "node -> incoming/outgoing artificial nodes (agg of parallel edges) -> node". The current version does not quite emphasize the technical challenges of such adoption.
> >
> > **Introduction of Artificial Nodes:**
> > The novelty and necessity of utilizing artificial nodes in MEGA-GNN would also benefit greatly from ablation studies that could illustrate their impact on performance. Such empirical evidence would enhance understanding of their role in the multigraph context.
> >
> > **Impact of Ego-IDs:**
> > Given the distinctions between MEGA-GNN and Multi-GNN, studying the effect of Ego-IDs, specifically in this case, would provide valuable insights into their effects and interaction with other modifications.
> >
> > I encourage the author to incorporate the above discussion into your manuscript to clearly articulate the contributions specific to multigraphs and differentiate MEGA-GNN from existing solutions. Additional experiments, especially concerning the points mentioned above, would greatly strengthen your work.

---

> ### Author Response · Authors · 2024-11-28
>
> Thank you for your suggestions, which we tried to address as much as possible in our revision.
>
> The reviewer will agree that showing the connections with the existing work, even if these connections are very remote, is part of the scientific writing process. In fact, the forward and the reverse edges can be considered as two different types of relationships, but that is only a very small part of our story. If one simply uses a relational GNN (e.g. R-GCN) over a multigraph without making multigraph-specific adjustments to the model, such as the ones we presented in Section 3 or the ones presented in the Multi-GNN paper, the results will be very poor. In fact, the Multi-GNN paper had already evaluated R-GCN and shown that it was not a competitive baseline when using the same multigraph datasets. That is why we did not explore this approach any further in our work.

---

> > ### Author Response · Authors · 2024-12-01
> > **Ego-IDs**
> >
> > Final clarification about EGO-IDs: although the ablation results we have provided in Table 4 show that the use of Ego-IDs does not lead to consistent accuracy improvements, we use EGO-IDs in the universality proof (see Appendix A,3, Proof of Lemma 3.1).   Namely, without defining an ego-node (a starting point), it would not be possible to assign unique ids to the remaining nodes.
> >
> > We hope that we have been able to address most of the concerns of the reviewer through our revision and responses.

---

### Author Response · Authors · 2024-11-27
**Summary of Revisions**

Dear reviewers,

Thank you for your feedback and suggestions. We have significantly revised our submission based on your comments. We have made the following important changes in our revised submission:

**Section 3.2**
- We have improved our notation to fully explain the way multigraph edge features are used both by standard GNNs and our multigraph GNNs.
- We have included formal definitions of universality, permutation equivariance, and strict total order.

**Sections 3.3 and 3.4**
- We have updated these sections to use the new notation in addition to simplifying their description.

**Section 3.5**
- We have significantly re-written this section to better explain our motivation for such a theoretical analysis and the most important insights we have derived:
- Both MEGA-GNN and Multi-GNN are permutation equivariant and universal when there is a strict total ordering of the edges.
- When such an ordering is not available, 1) MEGA-GNN is permutation equivariant but non-universal, and 2) Multi-GNN is universal but not permutation equivariant.

**Section 4.3**
- This is a new section that includes ablation studies of MEGA-GNN as well as inference throughput comparisons with the state-of-the-art baseline (Multi-GNN)
- The ablation studies show that the single most important factor contributing to MEGA-GNN’s high accuracy is the two-stage edge-aggregation mechanism taking place in MEGA-GNN’s message passing layers. Bi-directional message passing enables more moderate accuracy gains, and EGO-IDs do not lead to any consistent accuracy improvements.
- Even the most basic MEGA-GNN variant is significantly more accurate than ADAMM, proving the importance of performing multi-edge aggregations within the message passing layers rather than applying only a single multi-edge aggregation in a pre-embedding layer.
- Our throughput analysis shows that MEGA-GNN is only slightly slower than Multi-GNN. Furthermore, MEGA-GNN becomes significantly faster when bi-directional message passing is disabled.

**Section 5**
- We have included a new paragraph at the end of this section to discuss the relationships and differences between our work, hypergraph GNNs, and relational GNNs.

**Appendix D**
- We have added a brand new Appendix D Section that covers the computational and memory overheads of supporting multi-edge aggregations as well as some performance optimisation opportunities.

**Appendix E**
- We have added Appendix E, which includes additional experiments conducted on the baseline method, Multi-GNN. Specifically, we investigate the impact of edge permutations.

---

### Author Response · Authors · 2024-12-03
**Final remarks**

Dear reviewers,

Thank you again for all your constructive comments.

In your final evaluations, we would be glad if you could take into account our revisions as well as our following significant results:
- On IBM's extremely imbalanced Anti-Money Laundering datasets, our MEGA-GNNs outperform the state-of-the-art multi-graph neural network solution (Multi-GNN) by up to 13% in terms of the minority class F1 score.
- On a real-world phishing classification dataset, our MEGA-GNNs outperform the only other permutation-invariant multi-graph GNN model (ADAMM) by 30%.
- Our MEGA-GNNs are not only permutation invariant, but also universal when the graph edges are contextually (e.g., temporally) ordered.

---

### Meta-Review · Area_Chair_iGuf · 2024-12-21

**Metareview:**

**(a) Scientific Claims and Findings:**
The paper introduces MEGA-GNN, a novel framework designed to enhance Graph Neural Networks (GNNs) for multigraphs—graphs that permit multiple parallel edges between node pairs. Traditional GNNs often aggregate messages from all connected edges simultaneously, potentially overlooking the distinct contributions of parallel edges. MEGA-GNN addresses this by implementing a two-stage aggregation process: initially aggregating messages from parallel edges, followed by node-level aggregation from distinct neighbors. This design ensures permutation equivariance and invariance, maintaining the multigraph's topological structure. The authors demonstrate that MEGA-GNN achieves universality given a strict total order on edges. Empirical evaluations on synthetic and real-world financial transaction datasets indicate that MEGA-GNN either outperforms or matches the accuracy of current state-of-the-art models.

**(b) Strengths:**
* Innovative Aggregation Mechanism: MEGA-GNN's two-stage aggregation process effectively distinguishes between parallel edges and distinct neighbors, enhancing the expressive power of GNNs in multigraph contexts.
* Theoretical Guarantees: The framework upholds permutation equivariance and invariance properties, ensuring consistent performance across varying multigraph structures.
* Empirical Performance: The model demonstrates superior or comparable accuracy to existing solutions on both synthetic and real-world datasets, particularly in financial fraud detection scenarios.

**(c) Weaknesses:**
* Scalability Concerns: The paper lacks a comprehensive analysis of MEGA-GNN's scalability, especially when applied to large-scale multigraphs prevalent in practical applications.
* Computational Complexity: The introduction of a bi-directional multi-edge aggregation mechanism may increase computational demands, potentially limiting the model's efficiency in large datasets.
* Generality of Applicability: While the model excels in financial transaction datasets, its performance across diverse domains remains unexplored, raising questions about its generalizability.

**(d) Reasons for Rejection:**
After careful consideration, the decision to reject the paper is based on the following factors:
1. Insufficient Scalability Analysis: While a new appendix section (D) has been added during the rebuttal to provide a theoretical analysis of the algorithms complexity, no run-time tables were provided. This leaves uncertainty regarding MEGA-GNN's applicability to large-scale multigraphs.
2. Potential Computational Overhead: The increased complexity introduced by the bi-directional multi-edge aggregation mechanism may hinder the model's efficiency, especially in extensive datasets, without sufficient justification or optimization strategies.
3. Limited Domain Evaluation: The model's evaluation is primarily confined to financial transaction datasets, lacking evidence of its effectiveness across a broader range of applications, which limits the assessment of its generalizability.
Addressing these concerns through comprehensive scalability analyses, optimization of computational efficiency, and validation across diverse domains would significantly strengthen the paper's contributions and its potential for acceptance in future submissions.

**Additional Comments On Reviewer Discussion:**

Two reviewers, Reviewer Q2cP and Euww, remain convinced that the works is not ready for publication or would at least warrant another round of reviews.

While a misunderstanding regarding hypergraphs and multigraphs could be addressed during the rebuttal, Reviewer Q2cP remains doubtful about the novelty of ideas or significance of contribution.

Reviewer Euww found the method to be complicated and the insights resulting from the theoretical statements limited. The fact that a method is complicated can be justified by significant performance gains. Yet, the challenge is to communicate the benefits and potential intuition behind it and its novel contributions effectively. These can be evaluated in another round of review that judges the revised paper in its entirety.

---

### Decision · Program_Chairs · 2025-01-22

Reject